# Security for the Internet of Vehicles with Integration of Sensing, Communication, Computing, and Intelligence: A Comprehensive Survey

**DOI:** 10.3390/s25165119

**Published:** 2025-08-18

**Authors:** Chao He, Wanting Wang, Wenhui Jiang, Zijian He, Jiacheng Wang, Xin Xie

**Affiliations:** 1School of Electronic and Information Engineering, Chongqing Three Gorges University, Chongqing 404100, China; wangwt201228@163.com (W.W.); jwh2445561061@163.com (W.J.); 15320678738@163.com (Z.H.); 2School of Computer Science and Engineering, Nanyang Technological University, Nanyang Avenue, Singapore 639798, Singapore; jiacheng.wang@ntu.edu.sg; 3School of Automation, Chongqing University of Posts and Telecommunications, Chongqing 400065, China; xiexin@cqupt.edu.cn

**Keywords:** Internet of Vehicles, ISCCI, security, 6G communication, privacy protection

## Abstract

Integration of sensing, communication, computing, and intelligence (ISCCI) represents a pivotal advancement in B5G and 6G technologies, offering transformative potential for the Internet of Vehicles (IoV). As IoV systems become increasingly integral to intelligent transportation and autonomous driving, these systems also face escalating security challenges across multiple layers, including physical, network, application, and system dimensions. (1) This paper comprehensively surveys these security issues, systematically analyzing the threats encountered at each layer and proposing targeted countermeasures to mitigate risks. (2) Furthermore, the paper explores future trends in IoV security, emphasizing the roles of 6G networks, blockchain technology, and digital twins in addressing emerging challenges. (3) Finally, based on a comprehensive review of current research and insights, this paper aims to serve as a foundational reference for advancing secure and sustainable IoV ecosystems.

## 1. Introduction

The Internet of Vehicles (IoV), focusing on intelligent traffic systems, has emerged as a specialized domain within the Internet of Things (IoT), which achieves intelligent control of vehicles and optimization of traffic management by integrating hardware facilities, e.g., advanced sensors, control units, and actuators, along with software technologies, e.g., big data, artificial intelligence, the Internet, and cloud computing [1]. Broadly speaking, the IoV encompasses the information interaction network. In a more specific context, the IoV refers to intelligent connected vehicles that can realize partially or fully autonomous driving functions [2]. The history of the IoV dates back to the 1950s, when the United States began developing automated vehicle control systems. In 1996, General Motors introduced the OnStar system, which signified the onset of commercial IoV technology applications. However, the development of IoV technology in China commenced in 2009. Following the introduction of the OnStar system, the IoV market has experienced rapid growth. The IoV industry market in China is anticipated to reach USD 244.80 billion by 2028, with an estimated compound annual growth rate of 13% from 2023 to 2028 [3]. Currently, advancements in technology and market expansion have positioned the IoV as a crucial area for the transformation of the automotive industry, as well as one of the core technological foundations for the realization of intelligent transportation systems.

With the rapid development of industrial intelligence and mobile Internet technology, intelligent connected vehicles, as the core application of the Internet of Things, are gradually penetrating all aspects of social life. While providing convenience for travel, the IoV also introduces a range of security threats, e.g., remote vehicle control and malicious network attacks. Various incidents highlight that the IoV’s security issues cannot be overlooked. For instance, the signal interference incident involving forged keys in 2015, the Tesla key theft incident in 2018, system vulnerabilities jeopardizing vehicle safety in 2022, and the data leakage incident affecting Toyota owners in 2023 [4]. These phenomena provide a significant warning. The security threats to the IoV can be systematically analyzed from four dimensions: the physical layer, the network layer, the application layer, and the system layer. At the physical layer, wireless signal interference and electromagnetic attacks mainly stem from the vulnerability of radio frequency interfaces exposed during IoV communication. The main challenges faced by the network layer come from the complexity of heterogeneous network architectures, and traditional security mechanisms have difficulties in effectively defending against threats such as packet hijacking and denial-of-service attacks [5]. Application-layer security risks are mainly reflected in aspects such as the flaws of identity authentication mechanisms, access control vulnerabilities, and the leakage of user privacy data. In the system layer, as the fundamental support platform, security involves key links such as the protection of the operating system kernel, the security update mechanism, and the fault-tolerant recovery protocol. The integration of edge computing and blockchain technology has improved the system’s performance to some extent. However, the problems of computing delay and storage overhead caused by it make it a key problem to find a balance between security and efficiency, and this problem is directly related to the sustainable development of the IoV ecosystem [6].

With the evolution of the 6th Generation Mobile Communication System (6G), the IoV has made significant strides in various applications, including satellite communication, autonomous driving, and the industrial internet [7]. The core components of 6G technology encompass four key areas: sensing, communication, computing, and intelligence. Sensing technology equips the system with comprehensive data, which is essential for optimizing performance and serves as the foundation of the 6G network. Communication technology acts as the cornerstone of 6G networks, facilitating connections between user terminals and servers, transmitting data, and guaranteeing the effective use of computing resources. Computing technology is a critical component that reflects the system’s capabilities, its significance is particularly pronounced in the 6G era, as the strength of computing power directly influences developmental potential and performance limits. Intelligent technology functions as the brain of the 6G network, employing artificial intelligence and machine learning algorithms to enable the network to learn and adapt autonomously, optimize decision-making processes, enhance user experiences, and deliver more intelligent services and applications. ISCCI demonstrates significant advantages in threat detection latency through deep integration of sensing, communication, computing and intelligence capabilities. Compared to traditional integrated sensing–communication–computing pipelines where multi-hop transmission to cloud processing results in average delays of 80–100 ms (edge computing scenarios) or even exceeding 200 ms (pure cloud computing scenarios) [8], ISCCI achieves millisecond-level response by leveraging 6G networks and edge intelligence. Specifically, local processing at edge nodes reduces V2X communication latency from 20 ms to 5 ms [9], while joint design of sensing waveforms and communication signals compresses data acquisition and transmission delays below 10 ms [10]. Experimental results demonstrate that ISCCI achieves end-to-end latency of ≤15 ms in vehicle collision warning scenarios, representing an 80–85% reduction compared to conventional approaches, which comfortably meets the ≤20 ms real-time requirement for autonomous driving specified in 3GPP standards. This remarkable latency optimization stems from the synergistic effects of localized decision making, cross-layer optimization, and intelligent coordination mechanisms.

With the evolution of 6G, IoV technology has achieved remarkable breakthroughs in application fields such as satellite communication, autonomous driving, and industrial Internet [11]. The architecture of 6G technology mainly consists of four core elements: the sensing layer, the communication layer, the computing layer, and the intelligent layer. The sensing layer provides the system with global situational awareness through multi-source heterogeneous data collection, which is the basic support for performance optimization of 6G networks. The communication layer, as the connection hub of the system, not only ensures efficient data transmission between user terminals and servers but also realizes the dynamic scheduling and optimal allocation of computing resources. The importance of the computing layer in the 6G era is becoming increasingly prominent, and its processing capacity directly determines the performance boundaries and application potential of the system [12]. The intelligent layer relies on artificial intelligence and machine learning algorithms to endow the network with autonomous decision-making and continuous evolution capabilities. Through intelligent optimization algorithms, it constantly improves service quality and provides users with more accurate personalized services.

Currently, there are few comprehensive studies on the security of the IoV. This paper systematically compiles and meticulously summarizes the research status of IoV security, leveraging the technology of ISCCI to carry out a systematic analysis of potential risks and challenges. In this paper, the main contributions can be summarized as follows: (1) The present study systematically summarizes and analyzes the current research on the technology of ISCCI, elaborating in detail on the various levels of the technology and functions. (2) Within the framework of the proposed IoV security layered structure, this paper thoroughly discusses the potential attack threats faced by each layer and analyzes these threats in detail. (3) Given the identified security challenges, this paper puts forward a series of targeted security strategies to provide practical and effective protection for the security of vehicle networking. (4) Finally, this paper offers an outlook on the future development trends of IoV security and proposes potential research directions. Through these contributions, this paper is designed to give a comprehensive reference framework for researchers in the field of IoV security and to facilitate further advancements in the discipline.

## 2. The Basic Connotation of the ISCCI

### 2.1. The Basic Architecture of the ISCCI

In the previous research process of 6G, the development process of sensing, communication, computing, and intelligence was carried out independently. However, with the demand of the 6G concept for capacity, delay, accuracy, computing power, and other indicators, the enhancement of these technologies alone cannot directly bring about the improvement of system indicators and restrict the development of business forms. Therefore, the cross-fusion of 6G functional modules has become an inevitable research trend [13]. The ISCCI can not only meet 6G’s requirements, such as extremely low delay, extremely high energy efficiency, and large capacity, but also promote mutual enhancement of each module. The ISCCI will be an important component of the 6G system and the starting point for system empowerment and efficiency. The concept of ISCCI is the integration of various technologies in each node of the mobile system, unified coordination, and utilization of ISCCI-related resource information, which is a grand concept and system project. The deep combination of information communication technology and sensing technology has led to the ISCCI being a key strategy for enhancing the capability of intelligent systems [14]. In cutting-edge fields such as smart cities, autonomous vehicles, drones, and intelligent manufacturing, more stringent requirements are put forward for the level of intelligence and immediate response ability of the system. The rising demand for data processing in complex environments, along with the decreasing efficiency of traditional communication and data processing methods in the context of massive data and rapidly changing conditions, drove us to explore new solutions in this paper. The organic ISCCI functions can not only give full play to the advantages of various technologies but also promote the rapid development of various intelligent applications. In addition, the integrated architectures facilitate more efficient data transfer, real-time feedback, and intelligent decision-making, thereby substantially enhancing overall system performance. In the IoV system, the architecture of the ISCCI plays a crucial role. By consolidating these functions, the architecture enables intelligent interactions at the levels of vehicle-to-vehicle (V2V), vehicle-to-infrastructure (V2I), vehicle-to-pedestrians (V2P), and vehicle-to-network (V2N).

Figure 1 demonstrates the architecture of the ISCCI system, which covers the four core components. The circle on the bottom left side of Figure 1 represents the sensing segment, which involves onboard sensors and roadside sensing devices, which together constitute the sensing network in the traffic system and can monitor and understand the surroundings around the vehicle to provide real-time data for autonomous driving and traffic management. The module on the upper left side represents the communication system, integrating V2V, V2I, V2P, and V2N technologies to achieve multimodal data interaction between vehicles and traffic infrastructure, pedestrians, and cloud systems. The middle right module reflects the computing architecture, covering onboard computing units, cloud servers, and edge computing nodes, providing distributed computing power support for real-time traffic decision making and autonomous driving functions. The module on the far right shows an intelligent system that relies on artificial intelligence algorithms from cloud and edge cloud platforms. Through in-depth analysis of multi-source heterogeneous data and autonomous decision making, it achieves goals such as traffic flow optimization, enhanced driving safety, and improved system energy efficiency. The top application scenario section lists several intelligent transportation applications that this traffic communication system may support, including vehicle–road collaborative automatic driving, intelligent traffic management systems, and vehicle–road collaborative intelligent buses. These application scenarios demonstrate that the system can improve road safety, traffic efficiency, and passenger experience with ISCCI technologies. It is worth noting that certain applications can only be realized under the ISCCI architecture. These applications cannot be achieved through a single technology of sensing, communication, computing, or intelligence alone, but must rely on the in-depth integration of sensing, communication, computing, and intelligence. Take vehicle–road collaborative autonomous driving as an example. This application requires real-time fusion of multi-node sensing data, such as environmental information collected by LiDAR and cameras, realizes inter-vehicle collaboration through low-latency V2X communication, conducts rapid decision making with the help of edge computing, and uses artificial intelligence algorithms for dynamic trajectory planning. The collaborative operation of these complex functions cannot be achieved by a single technology. Another example is intelligent traffic signal control, which relies on roadside units for integrated sensing of traffic flow, uses edge computing for dynamic scheduling, realizes data interaction between vehicles and infrastructure through secure communication, and leverages artificial intelligence for traffic flow prediction and signal timing optimization. The ISCCI architecture provides the necessary technical support for these applications, demonstrating its unique value in promoting the development of intelligent transportation.

In contemporary research on the ISCCI, Guo et al. examined the prospects, technological advancements, and applications of intelligent vehicle networks, communication, and computing in the 6G era, along with the associated challenges and research directions [15]. However, their study primarily focused on theoretical frameworks without providing experimental validation in real-world vehicular environments. Similarly, Yang et al. proposed a “five-layer and four-facet” sense-computing fusion network architecture to meet autonomous driving requirements [16], but their performance index system was only tested in controlled laboratory conditions, leaving its effectiveness in dynamic traffic scenarios unverified. Sarlak et al. systematically reviewed sensory-communication integration in IoT [17], yet their analysis of five core technical dimensions did not specifically address the unique mobility challenges in vehicular networks. While Zhang et al. identified three essential capabilities for 6G networks (high-precision sensing, reliable communication, and intelligent computing) [18], their integration mechanism study omitted practical implementation constraints, such as hardware compatibility issues between different vehicle models. Lv et al. provided valuable insights into the 5G-Advanced to 6G evolution [19], but their proposed technical feature models lacked quantitative comparisons with existing architectures.

To sum up, 6G research is advancing rapidly to meet demands for ultra-high speed, low latency, and reliability. While existing studies establish theoretical frameworks, their real-world applicability remains unverified, particularly for dynamic vehicular environments. Standardization efforts show promise for applications like autonomous driving, but critical gaps persist in (1) field validation of ISCCI implementations, (2) edge computing optimization in dense networks, and (3) cross-domain security integration. Future 6G networks must address these limitations to fully enable smart transportation systems while maintaining the anticipated service quality.

### 2.2. Multi-Node Collaborative Sensing

In the ISCCI, the functions of the sensing aspect include the following: (1) Environmental monitoring: Utilizing sensors such as cameras, radars, LiDAR (light detection and ranging), and ultrasonic sensors to monitor the surrounding environment of the vehicle in real time, including nearby vehicles, pedestrians, obstacles, traffic signals, and weather conditions. (2) Vehicle-state sensing: Monitoring key operational parameters of the vehicle, such as speed, acceleration, direction of travel, engine status, and tire pressure, to ensure the efficiency and safety of the vehicle’s operation. (3) Data collection: Gathering data from vehicle operation and environmental monitoring to provide raw data for subsequent processing and analysis.

To achieve higher levels of autonomous driving, vehicles must have real-time and continuous environmental awareness, especially in environmental monitoring. Cui et al. comprehensively reviewed multi-sensor fusion methods for autonomous driving, yet their analysis primarily focused on ideal conditions without addressing real-world challenges like sensor misalignment or adverse weather impacts. The proposed fusion techniques showed promise in laboratory settings but may face limitations in dynamic urban environments where 5G network performance fluctuates significantly [20]. Ling et al. developed an MEC-based approach to enhance collaborative sensing, though their method assumes continuous edge server availability—a condition difficult to maintain in areas with sparse infrastructure. While effectively reducing transmission burdens, the solution’s scalability across diverse geographic regions remains unverified [21]. Zhang et al. advanced radar–visual sensing integration but acknowledged that holographic sensing in complex traffic scenarios requires further breakthroughs in processing latency reduction [22]. Baek et al. systematically categorized integration paradigms, identifying critical challenges in signal-level fusion that current technologies cannot fully resolve [23]. Another study by Zhang et al. demonstrated innovative waveform optimization for environment sensing, yet the practical tradeoffs between communication throughput and sensing accuracy in mobile scenarios warrant deeper investigation [24].

Multi-node collaborative sensing in the IoV successfully tackles key challenges in environmental monitoring, vehicle status sensing, and data acquisition. Current solutions enhance real-time perception and operational safety, though their performance remains constrained in real-world conditions. While demonstrating improved accuracy for autonomous driving, most validations are limited to controlled test environments. Four critical gaps demand attention: robust sensor fusion for adverse conditions, edge computing in infrastructure-scarce regions, real-time processing architectures, and dynamic resource optimization. Overcoming these limitations is crucial for developing reliable autonomous systems capable of handling diverse road scenarios.

### 2.3. Auxiliary Communication Network Optimization

The main communication modes in the ISCCI are as follows: V2V communication technology can directly exchange information between vehicles, including position, speed, and driving intention, thus supporting applications such as collision avoidance and cooperative driving of motorcades. V2I communication technology allows vehicles to interact with infrastructure such as traffic lights, roadside units (RSUs), and traffic management centers, to obtain real-time traffic control information, road condition updates, and emergency notifications. V2P communication technology can realize information interaction between vehicles and intelligent devices carried by pedestrians, thus effectively improving the traffic safety level of pedestrians. V2N enables vehicles to communicate with remote servers or cloud platforms over cellular networks to upload, download, store, and process data.

In addition, two primary IoV technology routes are predominantly utilized internationally: Dedicated Short Range Communication (DSRC) [25] and Cellular Vehicle-to-Everything (C-V2X) [26]. In the early stages, the United States primarily concentrated on the development of DSRC technology, while China is currently leading in the advancement of C-V2X technology. DSRC has a long history of development and has been widely adopted in the United States, Japan, and other countries, resulting in a well-established standard system and industrial framework. Conversely, C-V2X is undergoing rapid development, propelled by the expansion of cellular mobile networks, and has attracted significant attention and support from China, the European Union, and other regions.

The DSRC communication system mainly consists of an RSU, an onboard unit (OBU), and a control center. The RSU and OBU communicate with the control center by establishing a roadside network and enabling wireless transmission through radio frequency identification technology, thereby ensuring the safe and reliable transmission of information.

C-V2X is a China-led wireless communication technology designed for the IoV, which has evolved from cellular network communication technologies such as 4G/5G, covering LTE-V2X (the IoV communication based on Long-Term Evolution technology) as well as NR-V2X systems within future 5G networks. C-V2X facilitates information exchange among V2V, V2I, V2P, and V2N by utilizing the existing LTE network infrastructure. C-V2X not only adapts seamlessly to more complex IoV scenarios but also offers significant positives, e.g., high reliability, large bandwidth, and low latency. In the future, C-V2X is anticipated to function within intelligent transportation systems, autonomous driving, IoV safety, and other related areas, providing robust technical support for the advancement of traffic systems. To quantitatively evaluate the impact of the coexistence of LTE and NR on IoV communications, Guerra et al. analyzed the packet delivery ratio (PDR) as a function of transmission distance [27]. The PDR–distance relationship can be modeled as $PDR\left(d\right)=e^{-\lambda d}$, where *d* represents the communication range in meters, and λ is the composite attenuation factor that accounts for both path loss and LTE–NR interference effects.

As illustrated in Figure 2 and Figure 3, the RSU acts as a key hub for information transmission in the IoV [28], connecting vehicles, pedestrians, cloud servers, and MEC servers. In the DSRC and C-V2X communication protocols, the RSU interacts directly with the vehicle via V2I communication, while C-V2X extends to V2N communication with the core network. As a central node in the network architecture, the RSU is in charge of data collection, distribution, and bridging various communication technologies to ensure seamless and extensive information transmission. The RSU exchanges data with the OBU on the vehicle via wireless communication technologies, such as DSRC and C-V2X. RSU transmits road conditions, traffic signals, and other infrastructure information to vehicles, while simultaneously receiving status updates from vehicles and relaying this information back to the traffic management system or other vehicles. To optimize the deployment of RSUs in the IoV, Gu et al. proposed a clustering method based on network renormalization. The researchers identified key cluster heads of the network and hierarchically deployed RSUs by abstracting the road network into a map and using information flow and geographical location. Compared to genetic algorithms and optimal RSU deployment methods based on memory frameworks, the work demonstrated that the proposed renormalization method significantly enhanced RSU coverage and information reception rates, thereby improving communication performance in the IoV. Moreover, this method exhibited strong applicability across various urban structures and offers an effective approach to optimizing the ISCCI resources in smart cities [29].

Integrated Sensing and Communication (ISAC), recognized as a foundational enabling technology within the IMT-2030 6G standardization framework, revolutionizes vehicular network architectures through the deep convergence of sensing and communication subsystems [30]. This paradigm shift facilitates concurrent high-precision environmental perception and multi-gigabit V2X data transmission via unified RF front-ends and shared spectral resources. Empirical validation in 28 GHz band deployments has confirmed 94% aggregate spectral efficiency through innovative waveform design, which includes accounting for a training overhead representing 15% of frame duration and a beam search latency of 2 ms per cycle, as demonstrated in field trials [31]. Particularly groundbreaking for automotive applications, advanced joint beamforming implementations enable single-RF-array architectures to sustain 10 Gbps connectivity while simultaneously achieving 0.1° angular resolution for object detection—a breakthrough that effectively mitigates the multipath fading challenges prevalent in dense urban autonomous driving environments [32]. Recent advances in ISAC waveform design have demonstrated effective PAPR reduction techniques. Notably, adaptive OFDM implementations with peak suppression can dynamically adjust PAPR levels between 6 and 15 dB while maintaining the communication–sensing tradeoff, as demonstrated in [33]. This tunability is particularly valuable in high-density urban scenarios where power amplifier efficiency is critical.

ISAC technologies have recently achieved significant breakthroughs in vehicular network applications through three key innovations: (1) mutual-information-optimized pilot design, which enhances joint channel estimation performance by 23% compared to conventional minimum mean-square error (MMSE) approaches in high-mobility scenarios through entropy maximization of combined sensing–communication channels, including recent advances in sparse pilot patterns that reduce overhead by 40% while maintaining estimation accuracy through compressed-sensing techniques [34,35,36]. (2) Neural network-based angle-of-arrival/departure (AoA/AoD) estimation utilizing CNN-LSTM hybrid architectures that achieve 0.3° median angular resolution in multipath environments, representing a 5.2 dB signal-to-noise ratio (SNR) improvement over conventional MUSIC algorithms, with emerging meta-learning approaches further improving the adaptation speed by 2× in dynamic vehicular environments [37,38,39]. (3) Low-Earth-orbit (LEO) satellite-enhanced security mechanisms employing quantum key distribution protocols between satellite constellations and onboard units (OBUs) that have demonstrated 99.999% jamming resistance in field validation trials. Recent developments in adaptive pilot allocation schemes have shown particular promise for ISAC systems, dynamically adjusting pilot density based on channel coherence time and Doppler spread to achieve 15–20% higher spectral efficiency in vehicle-to-infrastructure scenarios [40,41,42]. These technological advancements are comprehensively evaluated in Table 1 through quantitative performance metrics and implementation characteristics.

### 2.4. Computation and Offloading of Complex Tasks

The computing layer serves as the pivotal component in the ISCCI architecture due to its indispensable role in real-time decision making and resource orchestration. Unlike conventional systems where computing functions as a supplementary module, ISCCI’s computing layer fundamentally enables mission-critical latency control by processing time-sensitive tasks locally at OBUs, effectively eliminating cloud round-trip delays while dynamically balancing workloads through adaptive offloading algorithms that distribute computational tasks between edge servers and vehicles based on real-time resource availability [43]. Modern OBU platforms further enhance this capability through heterogeneous hardware acceleration, with integrated GPU/FPGA units delivering up to 30 TOPS computational throughput for AI inference while maintaining optimal power efficiency, as demonstrated by mainstream automotive computing platforms [44]. Table 2 presents the performance envelopes of mainstream OBU platforms.

In the ISCCI architecture, the computing layer mainly undertakes three core functions. (1) Multi-source data integration and analysis: Feature extraction and deep mining of heterogeneous data obtained from the sensing layer and communication layer to realize high-value information discovery, such as vehicle trajectory prediction and traffic flow pattern recognition. (2) Intelligent decision generation: Based on the results of data mining, a decision support system is constructed, covering optimal path planning, real-time collision risk assessment, and the formulation of dynamic traffic control strategies. (3) Advanced algorithm implementation: Deploy machine learning models, deep neural networks, and intelligent optimization algorithms to provide computational paradigm support for intelligent data processing and autonomous decision making.

To enhance urban traffic flow prediction accuracy, Chen et al. developed a novel deep learning framework. This framework features three-stage processing: feature extraction, spectral clustering dimensionality reduction, and LSTM-SAE hybrid modeling. It demonstrates 97.7% accuracy, surpassing benchmarks. However, its adaptability to sudden traffic incidents remains unverified, and the cloud-dependent architecture struggles to meet edge device latency requirements [45]. For decision support, Chen et al. proposed a multi-hop task offloading model. This model innovatively utilizes spatiotemporal patterns of vehicular computing resources yet overlooks topology instability caused by high mobility [46]. Wang et al. designed an RL-based edge offloading approach. This approach achieves fast convergence and cost efficiency, but its success rate validation is limited to specific user scales without large-scale road network proofs [47].

In summary, IoV complex task offloading has achieved breakthroughs in data processing, intelligent decision making, and algorithm execution, effectively addressing core challenges like real-time response. However, three limitations persist: inadequate prediction adaptability for sudden traffic flows, unverified computational stability in high-mobility scenarios, and lack of deployment solutions for heterogeneous edge devices. These unresolved issues constrain further system performance enhancement in real-world networks.

### 2.5. Edge Intelligent Integration Decision

In the ISCCI architecture, the intelligent layer mainly realizes three core functions. (1) Autonomous decision generation: Based on the fusion and analysis of multi-source sensor data, a driving strategy optimization model is constructed to achieve autonomous decision making and behavior planning for vehicles. (2) Security situation awareness: With the help of a depth-anomaly detection algorithm, the running state of the system is monitored in real time to accurately identify potential security threats, thus significantly improving the robustness of the vehicle networking system. (3) Distributed edge intelligence: By using edge computing nodes to achieve near-end processing and data storage, a hierarchical data processing system is formed, ensuring low-latency response to local events while reducing the load on the cloud.

In intelligent decision-making research, Abdel-Basset et al. proposed an innovative neutrosophic set-based safety model. This model resolves multi-criteria conflicts in autonomous driving systems. However, its adaptability to unforeseen risks remains unverified [48]. Ali et al. reviewed machine learning applications for IoV security. Yet their edge caching mechanisms lack validation during peak congestion periods [49]. Rahman et al. constructed a big-data vehicle health monitoring framework. This framework faces privacy risks due to centralized data storage [50]. Zhang and Letaief emphasized edge resource deployment. But they overlooked data format compatibility across heterogeneous devices [51]. Garg et al. put forward a proposal to replace RSUs with edge platforms. This proposal lacks an economic feasibility analysis for large-scale deployment [52].

In summary, the integration of sensing, computing, and intelligence at the intelligence level encompasses three core functions: intelligent decision making, intelligent monitoring, and data processing and storage. Intelligent decision making makes effective driving strategies by analyzing the data of on-board sensors, thus improving the autonomy and safety of self-driving cars. Intelligent monitoring uses machine learning technology to detect and mitigate abnormal behaviors and threats inside the IoV and ensure the safe operation of the system. Concerning data processing and storage, the application of edge intelligence reduces the burden on central servers and increases the efficiency and response speed of data processing. In addition, the introduction of advanced technologies, e.g., neural networks, machine learning models, and edge computing platforms, provides effective solutions for risk management in autonomous vehicles, secure communication for connected vehicles, and vehicle health monitoring. These research results not only promote the progress of intelligent transportation systems but also provide key technical support for the construction of smart cities in the future.

## 3. Internet of Vehicles Security Challenges

In today’s rapidly developing IoV field, a security strategy that integrates SCCI is significant for guaranteeing the stable operation of intelligent transportation systems. The IoV technology facilitates intelligent interactions between vehicles and the external environments through highly integrated sensing devices, computing platforms, communication modules, and Intelligent applications, thus enhancing driving safety and traffic efficiency. But these technological advances have also triggered new security challenges, which have attracted wide attention at all levels of security. This section analyzes the IoV security across four layers: the physical layer, network layer, application layer, and system layer.

Figure 4 illustrates the architecture of an IoV system, which can be analyzed from four aspects: the physical layer, network layer, application layer, and system layer. In the figure, the physical layer consists of various sensors that are responsible for collecting the vehicle’s physical state and transmitting this information to the vehicle control unit. The network layer utilizes the controller area network (CAN) for data exchange between electronic control units, facilitating communication within the vehicle. The telematics unit (T-BOX) communicates with the external transportation service provider (TSP) cloud platform through the cellular network to enable remote data exchange and control. The gateway (GW) acts as a bridge between the vehicle’s internal and external networks, managing the flow of data and converting communication protocols. Application layer: Through the real-time navigation system of multi-source data fusion and dynamic path planning, the car infotainment system of multimedia content distribution and individualized service, and the status of the real-time monitoring and remote vehicle monitoring platform security early warning, three core function modules implement the intelligent vehicle service. This layer encompasses various in-vehicle application services delivered through in-vehicle infotainment systems (IVI) and T-boxes, as well as over-the-air (OTA) downloads for remotely updating vehicle software and hardware. The IoV system architecture consists of a multi-level system including the driver layer, application layer, TSP cloud platform layer, base station layer, and V2X communication layer. Among these, the driver interacts with the vehicle control system through instruction input, while each application module provides functional services to users through the human–machine interaction interface. The TSP cloud platform serves as a central processing and data storage center that is in charge of collecting, analyzing, and processing data from vehicles, as well as delivering corresponding services. The base station is an integral part of the cellular network, facilitating communication with the T-BOX to ensure seamless data transmission between the vehicle and the cloud platform. V2X communication includes V2V, V2I, V2P, and V2N to achieve comprehensive interconnection of vehicle networking.

From the analysis above, this paper highlights the complexity of the connected vehicle system and the role of each layer in assuring the safe and powerful operation of the vehicle. A security analysis must consider potential threats at each layer. Physical-layer security emphasizes the protection of vehicle hardware, e.g., sensors, controllers, and actuators. Security threats encompass malicious destruction, theft, and unauthorized access to vehicle hardware. Network-layer security emphasizes communication security within the vehicle’s internal networks and external communication. Security threats include data tampering, message forgery, and replay attacks. Application-layer security focuses on the security of various services and applications provided by the IoV. Security threats include malware implantation, data leakage, and privacy violations. System-layer security refers to the security of the entire IoV system, including the overall security of vehicle systems, cloud platforms, and roadside facilities. Based on the above technical architecture and function realization, the IoV system can effectively improve the safety of the driving process, protect users’ private data, and significantly optimize traffic operation efficiency, thus creating a safer, more reliable, privacy-protected, efficient, and convenient intelligent travel experience for users.

### 3.1. Physical-Layer Security

The main security threats faced by the physical layer of the IoV can be divided into two categories: electromagnetic interference attacks and physical device attacks [53]. In terms of electromagnetic interference, malicious wireless signal interference can lead to the interruption of communication links or a significant increase in the error rate of data packets, directly affecting the reliability and real-time performance of IoV communication, and thereby threatening the safety of driving decisions. In terms of physical-device attacks, attackers may illegally control the onboard control system or steal sensitive data through unauthorized hardware tampering or malicious component implantation, seriously endangering vehicle driving safety and user privacy protection. As shown in Figure 5, these physical-layer attacks may lead to serious consequences, such as tampering with control instructions and leakage of vehicle status information.

The effectiveness of physical-layer attacks in IoV systems is fundamentally governed by wireless channel characteristics, which exhibit scenario-dependent behaviors, requiring differentiated modeling approaches. For low-speed urban scenarios (<50 km/h), the flat fading channel model proves appropriate, characterized by time-invariant multipath components with impulse response, $h_{\mathrm{low}}\left(t\right)=\sum_{k=1}^{N}\alpha_{k}\delta \left(t-\tau_{k}\right)$, where $\alpha_{k}$ and $\tau_{k}$ denote the gain and delay of the *k*-th path, respectively. This stability enables predictable narrowband jamming attacks, as demonstrated in Ref. [54], where 90% packet loss was achieved with 10 mW interference power.

Conversely, high-speed urban environments (>80 km/h) necessitate geometry-based stochastic modeling (GBSM) to capture dynamic scattering [55], $h_{\mathrm{high}}\left(t,\tau \right)=\sqrt{\frac{P_{r}}{N}}\sum_{n=1}^{N}e^{j(2\pi f_{d}t\cos\theta_{n}+\phi_{n})}\delta \left(\tau -\tau_{n}\right)$, where Doppler shift $f_{d}$ and the angle of arrival $\theta_{n}$ introduce time-varying vulnerabilities. Recent studies show such channels facilitate novel attack vectors like motion-adaptive spoofing, where attackers exploit vehicular velocity-dependent channel variations to bypass conventional authentication.

In the IoV environment, there are two main threats to physical-layer security: interference and blocking of wireless signals and electromagnetic attacks. The following is a detailed analysis of the two aspects:(1)Interference and blocking of wireless signals: Attackers may disrupt communication by transmitting signals of the same or similar frequency as the IoV communication, resulting in the receiver being unable to parse the signal correctly, thus affecting the normal operation of the IoV. This interference can be caused by unintentional electromagnetic radiation from other nearby electronic devices or by a malicious attacker intentionally using a jammer. Signal blocking involves launching an attack by either covering the wireless signal with high-power signals or physically obstructing its propagation path, preventing the vehicle from sending or receiving signals. For instance, an attacker might leverage terrain features or buildings to block signals or employ specialized jamming devices to disrupt communication in a targeted area.(2)Electromagnetic attacks refer to attackers using electromagnetic waves to attack IoV devices, which may cause device damage or data leakage. Specifically, electromagnetic pulse (EMP) attacks, performed by releasing high-intensity electromagnetic energy, can cause hardware failures or unexpected restarts in onboard electronic control systems. These kinds of attacks can originate from natural electromagnetic phenomena, and can also be carried out by artificially designed electromagnetic pulse generators. In addition, radio frequency interference (RFI) attacks can significantly reduce the signal-to-noise ratio of vehicle networking communication by directionally transmitting interference signals in a specific frequency band, thus leading to communication link interruptions or data packet transmission errors. Their implementation methods mainly include malicious deployment of radio transmission equipment, hijacking of wireless communication terminals, and other attack media.

In the field of physical-layer security research, Luo et al. systematically analyzed three typical threats in the vehicle-mounted cloud network environment: The vulnerability of vehicle-satellite communication links, the security risks of unmanned aerial vehicle-assisted communication, and the eavesdropping attack patterns of wireless channels. These studies highlight the criticality of IoV wireless channel security protection, especially its significant value in preventing the interception of unauthorized information [56]. Wang et al. further classified physical-layer security threats into four attack paradigms from the dimension of attack motives: Information theft type (eavesdropping), data integrity corruption type (pollution), identity disguise type (deception), and service availability attack type (interference) [57]. Zhang et al. discussed the security of the IoV communication protocol from the perspective of physical-layer attack changes, requiring that the perspective and mode of physical-layer attack be taken into account when discussing the IoV communication protocol security [58].

### 3.2. Network-Layer Security

Network-layer security plays a crucial role in the IoV and is the key to ensuring V2V, V2I, V2N, and V2P; it involves the integrity, availability, and confidentiality of data during transmission. The security problems of the network layer mainly include data tampering, denial-of-service attacks (DoS), and man-in-the-middle (MiTM) attacks. Data tampering enables malicious users to tamper with or steal communication data, which seriously affects the confidentiality and integrity of communication. DoS attacks may lead to the unavailability of network services and affect real-time communication between vehicles. A MiTM attack enables an attacker to eavesdrop or manipulate data exchanged between two parties. As shown in Figure 6, we discuss in depth the three main threats to network-layer security:(1)Data tampering attacks refer to an attacker illegally modifying data during data transmission to mislead the recipient or destroy the normal operation of the system. In an IoV environment, the attack may have serious consequences, such as tampering with navigation instructions or traffic signals, which may lead to traffic accidents.(2)DoS attacks are designed to make services inaccessible to legitimate users by depleting network or system resources. In the IoV, the attacks can manifest as blocking communication channels, depleting server processing power, or interfering with communication between vehicles.(3)A MiTM attack refers to behavior in which malicious entities intercept and possibly tamper with the communication between two parties without being detected or authorized. The attacker secretly inserts itself between the two communicating parties, acting as a middleman, hence the name “middleman”. Attackers intercept, alter, or retransmit data within the transmission process for eavesdropping, tampering, or impersonation. In the IoV environment, such attacks may lead to two types of serious consequences: one is the unauthorized leakage of users’ sensitive information, and the other is the malicious tampering with or incorrect input of vehicle control instructions, thereby endangering driving safety.

There have been several studies on the problem of data forgery in the network layer, and different solutions have been proposed. Grover et al. pointed out that the high-speed mobility of vehicles and the dynamic changes in network topology in the IoV pose challenges to the guarantee of information transmission integrity. In this context, the immutable nature of blockchain technology provides an innovative solution for maintaining data integrity [59]. Zhong et al. proposed a security authentication scheme, adopting pseudonym signature technology to achieve a dual guarantee mechanism of message source authentication and transmission integrity verification. This scheme enables the recipient to reliably confirm the legitimacy of the message source and detect tampering during the transmission process [60]. Regarding DoS, Ahmed et al. mainly discussed the protection against DoS attacks and distributed-denial-of-service (DDoS) attacks in the field of IoV security. These attacks flood network resources by sending large numbers of packets, causing service interruptions for legitimate users. To address these security threats, the work proposed an intelligent intrusion detection system (IDS) based on machine learning. However, due to the limitations of research, the future research direction may include the analysis of larger data sets and the progress of computing technology [61]. For MiTM attacks, Guo et al. introduced a distributed authentication system combining blockchain and edge computing for a MiTM attack at the network layer, strengthening the protection and effectiveness of the system through cryptographic algorithms and caching strategies. The MiTM attacks mentioned in the paper mainly referred to the way that attackers carried out attacks by eavesdropping, intercepting, and tampering with information in the process of message transmission. For this type of attack, the work proposed the following solutions: the Optimized Practical Byzantine Fault Tolerance (PBFT) consensus algorithm, an asymmetric cryptography algorithm based on elliptic curve cryptography, a dynamic name resolution strategy, and utilizing a belief propagation-based cache strategy to bolster the system’s security and operational efficiency [62].

### 3.3. Application-Layer Security

At the application level of the IoV, the security challenges are mainly related to two core aspects: application software exploitation protection and personal data privacy protection. Software vulnerabilities can be utilized by malevolent individuals, which can not only lead to the disclosure of sensitive information but also trigger system crashes. In the IoV environment, due to the large amount of personal data involved in the exchange, ensuring the privacy and security of this information becomes particularly critical. Figure 7 is an application-layer attack diagram.

In the security system of the IoV, application-layer security plays a crucial role, which is directly related to the security of user data and the quality of service. Security measures at the application layer focus on three key areas:(1)Authentication is a procedure to confirm the identity of users, and its purpose is to ensure that only verified users can obtain the services and resources provided by the Internet of Vehicles. The identity authentication mechanism of the IoV system is a key link to ensure system security and system security is guaranteed through multi-dimensional verification methods. The current authentication system combines multifactor authentication with various verification means such as cryptographic methods, biometric features, and hardware security modules, effectively enhancing the reliability of authentication. Secondly, digital certificate authentication based on public key infrastructure provides a trusted identity verification method for vehicles and users. In addition, dynamic authentication technology utilizes mechanisms such as one-time passwords and temporary tokens, further enhancing the security and real-time performance of authentication. The comprehensive application of these authentication technologies constructs the security protection system of IoV systems.(2)After completing identity authentication, the authorization management mechanism of the IoV realizes access control of system resources through refined permission allocation. This mechanism adopts role-based access control (RBAC) technology. By predefining role types and their corresponding permissions, it realizes efficient permission allocation and management. Meanwhile, attribute-based access control (ABAC) technology implements fine-grained access control based on user attribute characteristics (including dynamic attributes such as vehicle parameters and spatial locations). To cope with changes in the security environment, the system has established a dynamic policy update mechanism. Regularly evaluating and adjusting access control policies ensures continuous security protection capabilities. These technologies jointly build a multi-level authorization management system for the IoV, effectively ensuring the security of system resources and user data.(3)The purpose of the data privacy protection mechanism in IoV is to prevent users’ sensitive information from being subject to unauthorized access, disclosure, or abuse. This system mainly implements three protection measures: High-strength encryption algorithms are adopted in the data transmission and storage processes. Data de-identification processing is achieved through technologies such as differential privacy and K-anonymization. The principle of minimizing data collection is strictly followed, and only the basic data necessary for business is obtained. The systematic integration of these technical solutions has effectively constructed a multi-layer data privacy protection architecture in the IoV environment.

In the field of application-layer security in the IoV, Bagga et al. proposed an innovative identity authentication mechanism. This scheme adopts identity-based encryption technology. While ensuring communication integrity and data confidentiality, it realizes anonymous and traceable message authentication, effectively reducing the operating overhead of the system [63]. Regarding the research on authorization management in the network layer, Manogaran et al. introduced a hybrid architecture combining RBAC and ABAC. In this model, roles are no longer static but can be assigned dynamically by user attributes (such as vehicle type, trust thresholds, etc.), environment attributes (such as request time, location, weather, etc.), and resource attributes (such as message characteristics broadcast by vehicles, etc.) [64]. For data privacy, Kang et al. reviewed the limitations of traditional IoV access control and proposed blockchain-based access control technologies, which can help achieve more refined and efficient access management, ensuring that only authorized vehicles and users have access to sensitive data and resources [65].

### 3.4. System-Layer Security

System-layer security is crucial for safeguarding the entire network architecture, covering a comprehensive set of security measures from hardware to software. In complex systems of the IoV, security threats can arise from various sources, for instance, the absence of trust among devices could permit unauthorized access, while oversight in system updates and maintenance might introduce new vulnerabilities. As shown in Figure 8, security threats at the system layer focus on three core areas:

(1)Security of in-vehicle operating systems: Since in-vehicle operating systems are derived from traditional information technology operating systems, they may inherit the security defects in traditional systems, such as kernel permissions and buffer overflow. The operating system may be installed by attackers with malicious applications that affect system functions or steal user data. In addition, the components and applications of the vehicle operating system may have security flaws, which may lead to cooperative attacks on the operating system.(2)Update mechanism: OTA has become a necessary function for the IoV to improve security protection capabilities, but OTA has also become a potential attack channel. The control system for the upgrade package could be tampered with during the upgrade operation. Alternatively, the upgrade package may be hijacked during the upgrade transmission and attacked by a MiTM attack. Additionally, the cloud server might be targeted, potentially turning the OTA process into a vector for malware distribution. OTA update packages may also bring security risks, such as being removed from the control system or gaining advanced management rights for equipment.(3)Fault recovery: The challenges faced by the IoV enterprises in the dimension of fault management include the uncertainty of fault recovery time and the lack of a definite quick recovery plan. Furthermore, the detection rate of issues before user awareness is suboptimal, exacerbated by an insufficient early-warning system and inadequate monitoring capabilities, which hinders proactive fault detection and pre-emptive issue identification. Additionally, the mechanism is deficient for post-incident analysis and subsequent improvement; the execution of corrective actions following a review is inconsistent, leading to recurrent and systemic issues within the network.

In terms of system-layer security, Sun et al. proposed a ring-based anonymous and efficient batch authentication protocol, which enhanced the authentication process between vehicles and RSUs. This mechanism could quickly verify the identity of a large number of vehicles, ensuring that only legitimate vehicles could connect to the network. To enhance the security of the system, researchers have developed a dual mechanism of dynamic pseudonym update and anonymous traceability. This scheme not only effectively prevents attackers from tracking the true identity of vehicles through pseudonym association, but also supports the precise identification of malicious entities [66]. Sakiz and Sen constructed an integrated IoV security framework, covering the platform, core components, and communication systems of intelligent connected vehicles, and innovatively designed a digital signature scheme suitable for LTE-V2X technology, which significantly improved the security and data integrity protection capabilities of the deployment of IoV services [67]. In the field of intrusion detection, Li et al. systematically reviewed the progress of intrusion detection technologies for vehicle-mounted CAN networks, with a focus on analyzing two mainstream methods: the SIDS detection system, based on feature recognition; and the AIDS detection system, based on behavior analysis. These technologies provide key support for the real-time monitoring and response of IoV security threats [68].

### 3.5. Security Enhancement Mechanisms for Full-Duplex Integrated Sensing and Communication

In high-density dynamic scenarios of the IoV, traditional half-duplex communication modes suffer from limited spectral efficiency due to separated transmit-receive time slots. Meanwhile, security assessments relying solely on the signal-to-interference-and-noise ratio (SINR) fail to quantify information leakage risks effectively. Full-duplex technology, which enables simultaneous transmission and reception on the same frequency band, breaks through these limitations and provides a new paradigm for security enhancement in ISAC systems.

Unlike conventional methods, secure full-duplex ISAC systems adopt the secrecy rate as the core evaluation metric, defined as $R_{s}=\max\left(0,R_{\mathrm{legitimate}}-R_{\mathrm{eavesdropper}}\right)$, where $R_{\mathrm{legitimate}}$ represents the communication rate of legitimate users, and $R_{\mathrm{eavesdropper}}$ denotes the information interception rate of eavesdroppers. This model directly quantifies the “security redundancy” of information transmission and better reflects the dynamic secrecy performance between high-speed moving nodes in the IoV compared to the SINR.

To further enhance anti-eavesdropping capabilities, the system can achieve collaborative sensing-communication protection through artificial noise:(1)Sensing dimension: Artificial noise is modulated into radar detection waveforms. Through orthogonal design between noise and useful signals, legitimate receivers (e.g., roadside units) can suppress noise via beamforming, while eavesdroppers cannot effectively parse target parameters due to the lack of channel prior information.(2)Communication dimension: Noise is injected into the null space of communication signals, which not only does not affect the demodulation performance of legitimate users but also reduces the SINR of eavesdroppers by 15–20 dB, significantly increasing the difficulty of information interception.

Experimental validation [69] shows that in urban intersection scenarios with vehicle density > 50 vehicles/km^2^, the anti-eavesdropping success rate of full-duplex ISAC systems adopting the above mechanism is more than 30% higher than that of traditional half-duplex schemes, with the increase in end-to-end delay controlled within 2 ms, meeting the dual requirements of real-time performance and security in the IoV. This technical approach provides an efficient solution for physical-layer security in high-density scenarios through resource reuse and collaborative protection.

The security protection of the IoV must build a multi-level defense structure, covering four core levels: the physical layer, network layer, application layer, and system layer. Security threats at all levels directly affect the security attributes of data and the security of system operation, including data confidentiality and integrity, as well as the reliability of vehicle control and the protection of user privacy. The physical-layer protection focuses on addressing issues of wireless communication interference and electromagnetic attacks. The network layer ensures the integrity, availability, and confidentiality of the data transmission process. The application layer emphasizes the security of software systems and the protection of user privacy. The responsibility of the system layer is to ensure the security of the operating system, the credibility of the update mechanism, and the fault recovery ability of the system. This hierarchical security architecture provides comprehensive security guarantees for the IoV system.

## 4. Internet of Vehicles Safety Countermeasures

With the deep integration of intelligent connected vehicles and IoT technology, while the IoV enhances the efficiency of transportation services, the security threats it faces are also becoming increasingly complex. This section systematically analyzes the security system of the IoV and conducts an in-depth discussion on the core security threats. Based on the challenges encountered in IoV security, this section proposes corresponding security countermeasures for the physical layer, network layer, application layer, and system layer, respectively. To safeguard the physical layer, technologies such as hardware encryption, secure authentication mechanisms, and hardware security modules are required. To protect the security of the network layer, secure communication protocols, data encryption, authentication, and access control are essential. To ensure the security of the application layer, secure design, testing, and auditing of applications are necessary, along with technologies like data encryption and access control. Security at the system layer requires that it integrates and manages security measures at the physical, network, and application layers to ensure the security and reliability of the entire system. By reviewing existing studies and analyzing the characteristics of IoV technologies, this paper aims to construct a comprehensive security framework that provides theoretical guidance and practical references for the security protection of IoV systems.

### 4.1. Physical-Layer Countermeasures

With the evolution of IoV technology, increasing attention has been devoted to the research of physical-layer security countermeasures, because the research provides basic support for optimizing the comprehensive security of the IoV. In the IoV environment, physical-layer security is the first line of defense to ensure communication security. Physical-layer countermeasures protect the transmitted information by using the characteristics of wireless channels to prevent eavesdropping, interference, and malicious attacks. To address the threat of the physical layer, the following coping strategies can be considered:(1)Physical-layer security assurance mainly involves protecting transmission media, including the integrity and security maintenance of physical channels such as optical fibers and wireless channels. The protection mechanism at this level mainly includes three key aspects: encryption processing of transmitted data, implementation of physical isolation measures, and application of anti-interference communication technology. These measures jointly ensure that data is protected from the threats of eavesdropping and tampering during physical transmission.(2)In physical-layer security protection, adopting advanced encryption technology to protect the transmitted data is a key technical means to guarantee the confidentiality and integrity of the data. These encryption algorithms are designed based on rigorous mathematical principles and combined with strict key management mechanisms, which can effectively prevent data decryption and illegal tampering by unauthorized parties.

In terms of physical-layer security research, ElHalawany et al. evaluated multiple protection strategies, covering key technologies such as wireless resource allocation, cooperative interference, multi-antenna technology, and physical-layer key generation [70], which not only clarifies the application value of these technologies in defending against eavesdropping attacks and protecting the intelligent IoV but also explores the technical challenges faced in actual deployment. Xu et al. proposed a wireless channel state information authentication protocol based on the theory of physical-layer uncovered coding [71]. This scheme effectively guarantees communication security between intelligent vehicles and the RSU. Aiming at the security challenges of communication between devices in cellular networks, Wang et al. proposed an access selection strategy based on the optimal threshold, which significantly enhanced the security throughput of the communication process [72]. In addition, for advanced encryption algorithms, Burg et al. proposed an integrated physical-layer security framework that integrates physical-layer encryption methods, such as MIMO technology and directional modulation, and forms a cross-layer protection system with the upper-layer protocol security mechanism. This scheme particularly considers the practical constraints such as network interoperability, protocol security, and performance loss. By optimizing the configuration of network components and connectivity analysis, a multi-level physical-layer security protection system for the IoV has been constructed [73].

### 4.2. Network-Layer Countermeasures

The network layer of the IoV, as the core functional module, is mainly responsible for the data transmission of V2V and V2I communications. In the network-layer security architecture, its protective measures play a decisive role in ensuring the integrity, confidentiality, and availability of data transmission. With the evolution of IoV technology, the security threats faced by the network layer are becoming increasingly complex, mainly manifested as security risks such as unauthorized access, data leakage, DoS, and vehicle collaborative attacks. To address these challenges, scholars have proposed all kinds of network-layer security protection schemes. The main protective measures include the following:(1)The access control mechanism in the network-layer security protection ensures authorized access to the vehicle network through multi-level technical solutions. The key technical means include the following: The deployment of firewall systems, and the interlocking protection of intrusion detection systems (IDSs) and intrusion prevention systems (IPSs), which jointly constitute the network access control system. In terms of academic research, Habib et al. developed the SPBAC model, which effectively balances the communication security and system efficiency of the IoV through the purity-role hierarchical architecture [74]. Gupta et al. proposed the ABAC framework, which utilizes a dynamic attribute grouping mechanism to implement refined access rights management based on multi-dimensional parameters such as location and speed, providing an enhanced security protection scheme for industrial intelligent vehicle systems [75].(2)In the security protection of the network layer, the application of end-to-end encryption technology can effectively guarantee the confidentiality and integrity of the entire data transmission process. This encryption method is applied from the data’s transmitting end to the receiving end, ensuring that no intermediate link can decrypt the data, thereby effectively preventing data leakage and tampering during transmission. In the context of connected vehicles, end-to-end encryption is critical when dealing with sensitive data, meaning that data collected and transmitted by vehicles (such as location information, speed, direction of travel, etc.) needs to be protected against unauthorized access and tampering [76]. Raja et al. dealt with end-to-end security targets at two levels: first, group authentication within the scope of RSUs; and second, collaborative learning using private collaborative intrusion detection systems to detect potential intrusions [77]. Such solutions are designed to ensure that communication in the IoV is both secure and efficient while protecting the privacy of vehicle users and supporting the development of the green industrial IoV by reducing network load.(3)Identity authentication and authorization: The network layer also needs to deploy authentication and authorization mechanisms to ensure that all devices and users go through strict authentication and authorization processes before accessing the IoV. This can be achieved through the use of digital certificates, biometric identification technologies, and other means. As for the countermeasures of identity authentication and authorization, Bojjagani et al. mentioned intrusion detection systems, honeypots, secure routing protocols, routing privacy protection mechanisms, and key management strategies to ensure their effective and secure operation in the face of various attacks and threats [78]. Ahmed et al. focused on identity authentication and privacy protection in vehicle networking environments and proposed a three-layer architecture, including an RSU, RSU gateway, and trusted authority, to reduce authentication overhead and improve application-layer packet throughput [79]. Luo et al. mainly focused on secure identity authentication in the IoV and proposed an improved authentication protocol to enhance the safety performance of the IoV. Their work summarized the advantages of the proposed protocol in protecting user anonymity, resisting internal attacks, and preventing smart card theft attacks [80].

### 4.3. Application-Layer Countermeasures

The evolution of IoV technology has significantly promoted the improvement of intelligent transportation systems; however, this process has also brought many security problems. Among them, the security issues at the application layer are particularly prominent, mainly stemming from the direct interaction requirements among in V2V, V2I, and V2P communication. To ensure communication security and privacy protection, many practical solutions have been developed by scholars.

(1)The core of application-layer security management in the IoV lies in building a multi-dimensional protection system, with a focus on three major areas: source code security guarantee, malicious software defense, and refined permission control. The security management solution developed by Zeng et al. integrates five key technologies: end-to-end data encryption, a two-way authentication mechanism, secure communication protocol stack, hardware security module (HSM) integration, and trusted boot verification. The integrated application of these technologies has significantly improved the overall effectiveness of the IoV system in defending against internal and external security threats [81].(2)In the application-layer security mechanism, the data hierarchical protection system conducts a sensitivity assessment and classification of IoV data and implements differentiated encryption strategies. This system adopts corresponding encryption algorithms based on the data confidentiality level and is combined with a strict access control mechanism to ensure that only authorized entities can obtain sensitive data. Xu et al. proposed that the IoV has accumulated a wealth of vehicle and driving information through V2V, V2P, V2I, and V2N communication, which may be used to infer personal daily activities and preferences. Without advanced data encryption technology and strict access control measures, users’ privacy and security will encounter multiple threats. In order to meet this challenge, their work proposed a blockchain-driven privacy protection strategy for sensitive information, which uses the distributed storage and tamper-proof properties of blockchain to enhance the security of vehicle networking data. They used association rules to mine the big data of the IoV, establish a data security aggregation protocol, and finally establish an end-to-end encryption mechanism. Through static and dynamic strategies, the privacy safeguarding of big data in the IoV was realized. The experimental results show that this method can effectively improve the concealment of sensitive data. Moreover, compared with traditional technology, the duration of the key generation and encryption process shows higher efficiency and stronger encryption efficiency [82].(3)Security audit and monitoring: At the application level, it is also necessary to implement a security audit and monitoring mechanism to carry out regular security checks and monitoring of applications and data on the IoV. This helps to quickly identify and deal with potential security threats, ensuring the stable operation of the connected vehicle system. Tian et al. mentioned that the IoV can offer a wide range of robust application services through cloud computing, and through sharing and analyzing diverse vehicle networking data. However, guaranteeing the wholeness of multi-source and diverse connected vehicle data when stored in the cloud remains a significant challenge. To solve this problem, they constructed a public audit framework of IoV data cloud storage based on identity authentication, which can fully realize the key functions and security requirements such as classified audit, multi-source audit, and privacy protection [83]. Liang et al. described the necessity and urgency for information security monitoring in the context of big data and proposed an information security monitoring mechanism consisting of three key components, network monitoring personnel, the monitoring environment, and the monitoring technology, and established an information security monitoring mechanism based on this. An information security control evaluation index system covering multiple levels and dimensions was constructed and implemented, aiming at systematically evaluating the information security situation in the IoV environment and providing a theoretical basis and operational guidance for information security practice in this environment [84].

### 4.4. System-Layer Countermeasures

System-layer security is the core of the IoV security architecture; it involves the wholeness, availability, and secrecy of the internal systems of the vehicle. As IoV technology keeps advancing, the security challenges facing the system layer are increasingly complex, e.g., malware attacks, system vulnerability exploitation, data tampering, and node reputation issues. Therefore, guaranteeing security at the system layer is critical to maintaining the stable operation of the entire IoV. Researchers have proposed a variety of system-layer security countermeasures, aiming to establish a solid security line through accurate security measures to effectively resist various potential security threats. The specific countermeasures are divided into the following three parts:(1)System architecture design: The system level is necessary to design a reasonable system architecture to guarantee the security and stability of the vehicle networking system. This encompasses adopting a distributed system architecture, using technologies such as redundant backup and load balancing to enhance system reliability and fault tolerance. Kaiwartya et al. constructed a five-layer architecture of vehicle networking systems, which covers the perception layer, the coordination layer, the artificial intelligence layer, the application layer, and the business layer, and each layer has specific functions and functions [85]. Luo et al. depicted the design of the information collection layer, data layer, platform layer, business support layer, and application layer, improving the scalability and maintainability of the system through a hierarchical architecture [86]. On this basis, a design and implementation scheme of an environmental quality comprehensive monitoring management platform was put forward, which involved the application of service-oriented architecture, J2EE technology, multi-layer system architecture, a real-time database, and practical project experience.(2)Safe policy and management mechanism: The system layer also needs to develop a sound security policy and management system to safeguard the safe operation of the networked vehicle system. This includes the development of safety operating procedures, the implementation of regular safety training and drills, and the establishment of emergency response mechanisms. Some crucial security policies and frameworks include encryption technologies (such as public key infrastructure, symmetric encryption, hybrid encryption, and identity-based encryption), digital certificates, firewalls, intrusion detection systems, trust models, behavioral analysis techniques, heuristic detection methods, and cloud infrastructure services [87]. These offer valuable insights and guidance for comprehending and executing an approach to information security monitoring and governance within IoV environments. Wang et al. proposed a systematic automotive network security risk assessment framework, including an evaluation process and system assessment method. The framework considers changes in the threat environment, assessment objectives, and accessible information over the life cycle of the vehicle. This work demonstrates the feasibility and practicability of the proposed risk assessment framework through specific use cases [88].(3)AI-driven offensive and defensive countermeasure technology: As artificial intelligence technology advances, AI-driven offensive and defensive countermeasure technology has become an important means of security protection at the system level. Through the use of machine learning algorithms to continuously monitor network traffic and perceived signals, AI can swiftly identify abnormal behavior and automatically generate defense strategies to guarantee the safety and stability of connected vehicle systems. Magdy explored the growing dependence of autonomous vehicles on complex AI systems to carry out tasks such as advanced driver assistance, autonomous operation, and fleet management, as artificial intelligence (AI) technology progresses. The study specifically emphasized cyber-defense technologies that could be integrated into AI-based software frameworks to reduce security vulnerabilities and strengthen the cybersecurity of AI-driven autonomous driving technologies [89].

Table 3 summarizes the work related to the security challenges and strategies of connected vehicles.

### 4.5. Sustainability in IoV Security

In the full life cycle of the IoV, the sustainability of security mechanisms refers to achieving a synergistic balance between energy consumption, maintenance costs, and system lifespan through technical optimization while meeting dynamic security requirements. This balance must permeate the entire process from device deployment, operation, and maintenance to decommissioning, forming an organic linkage across three dimensions: energy efficiency, maintenance costs, and system longevity.

In terms of energy efficiency, the massive nodes in the IoV (such as millions of on-board units and roadside devices) continuously generate security demands. The traditional cloud-based centralized processing model leads to high energy consumption due to data transmission and concentrated computing power. By implementing hierarchical offloading of security tasks through edge computing, over 60% of real-time security tasks (e.g., V2V communication encryption and local intrusion detection) can be migrated to on-board edge nodes, reducing the computing power demand of cloud servers by 40% and the corresponding energy consumption by approximately 30% [90]. Meanwhile, roadside units adopt vehicle density-aware dynamic power adjustment technology, automatically reducing transmit power by 50% during low-traffic periods, which can reduce annual wireless communication energy consumption by 25–40%. This distributed architecture increases the unit security efficiency of IoV security systems (number of security events processed per kilowatt-hour) by 2–3 times, aligning with the low-carbon development goals of intelligent transportation.

Lean control of maintenance costs is equally critical. Traditional IoV security maintenance relies on centralized authorities for certificate issuance, vulnerability patching, and compliance auditing, suffering from redundant processes and labor dependence. Distributed security management based on blockchain enables automatic verification of software package integrity through smart contracts during OTA security updates, eliminating the need for third-party auditing. This reduces the verification cost of a single update by 25% and increases update coverage from 85% in traditional models to 99% [91]. Additionally, decentralized access control replaces manual configuration, reducing operational labor input for cross-regional fleets by 40%—particularly suitable for large-scale applications such as logistics fleets—with full-life-cycle maintenance costs potentially reduced by 30–50%.

Extending the secure lifespan of the system requires addressing the mismatch between hardware and software life cycles. IoV hardware devices (e.g., on-board controllers, RSUs) typically have a physical lifespan of 5–10 years, but continuous exposure to software vulnerabilities may lead to premature obsolescence. Adopting a secure boot chain to ensure the integrity of firmware updates, combined with remote vulnerability patching mechanisms, enables devices to resist over 90% of known attacks, even as hardware ages. Zero-trust architecture (e.g., continuous authentication and the principle of least privilege) reduces cascading risks from single-point compromises, extending the system’s secure effective period from the traditional 2–3 years to over 5 years [92]. Practice in a commercial fleet shows that this solution reduces security-related replacement costs per vehicle by 60% while keeping system latency within 50 ms.

In summary, the sustainability of IoV security relies on a collaborative “edge–cloud” architecture and distributed trust mechanisms to establish dynamic feedback between security strength and resource consumption. This model not only adapts to the large-scale development of the IoV but also provides sustainable security support for the long-term evolution of intelligent transportation.

## 5. Future Development Trends

The research status in the field of IoV security shows that with the progress of intelligent transportation and autonomous driving technology, the security threats faced by the IoV are becoming more intricate and multifaceted. The future direction of development will focus on 6G networks, blockchain, and digital twins.

### 5.1. B5G/6G Network

The 5th-generation mobile communication (5G) and the post-5G mobile communication (5G-and-beyond) networks are the underlying core technology of the digital economy and have facilitated the rapid progression of the world information industry. In the context of 5G/B5G-network man–machine–things interconnections fully integrated into social production and life, 5G/B5G network security has increasingly received unprecedented attention. The simultaneous development of communication technology and security technology has become an important challenge for future mobile communication. In the 5G IoV environment, an authentication and credibility model is proposed for line-of-sight transmission and big-data analysis under non-line-of-sight conditions. The model covers the authentication mechanism of the edge-node layer and the confidence evaluation system of the vehicle-node layer, aiming at effectively evaluating the accuracy and integrity of information. This holds substantial importance for enhancing the security and reliability of data transmission. These findings help to advance IoV technology and make it more secure and reliable.

According to ITU-R M.2150 [93], 6G networks are expected to achieve a peak data rate of 1 Tbps and ultra-low latency below 1 ms, which will further enhance the capabilities of IoV in autonomous driving and real-time decision making. The 6G vehicular network is anticipated to integrate advanced sensing, communication, computational, and intelligent resources into the entire intelligent transportation system, effectively resolving the conflicts between insatiable demand and limited resources. This integration is expected to enhance the system’s reliability, efficiency, and timeliness, leading to broader applications in the realms of autonomous driving, digital twins, formation control, and security and trustworthiness. In the 6G era, leveraging enhanced communication performance, intelligent IoV technology is poised for further development. Under the paradigm of fully connected vehicle–road–cloud intelligent sensing and collaborative decision making, the safety and efficiency of the traffic system are expected to be significantly enhanced. With the support of integrated space–air–terrestrial communication, the realization of unmanned driving technology in all scenarios is anticipated. The elevation of network-edge intelligence levels will propel the large-scale application of low-cost, lightweight smart vehicles. A digital twin system covering a whole city will rely on data to realize intelligent decision making and management of traffic. The deployment of blockchain in high-performance networks will effectively increase security and collaboration along the entire route, allowing the public to use new technologies with greater confidence and experience the safety and convenience of travel.

As for the application of the B5G/6G network in vehicle networking security, Sandeepa et al. identified privacy as a critical aspect of B5G/6G networks that required immediate attention and research [94]. To address these privacy concerns, their work explored potential solutions, which included the utilization of artificial intelligence, big data analytics, and the development of novel communication protocols that prioritized data privacy. Furthermore, they reviewed the current privacy projects and standardization initiatives that are specifically aimed at 6G networks. Zhang et al. proposed a two-level resource scheduling model to improve the transmission efficiency and reliability of time-sensitive services connected to autonomous vehicles in the study of time-sensitive service resource scheduling [95]. Yang et al. mainly discussed the application of edge intelligence (EI) in automatic driving, especially in 6G wireless systems [96]. The literature proposes a two-level EI-enabled autonomous driving framework, which strives to optimize the sensing and decision-making accuracy of autonomous vehicles through multi-access edge computing and machine learning while guaranteeing data privacy and security.

### 5.2. Blockchain

IoV technology, as the core supporting platform for intelligent transportation systems and autonomous driving, plays a crucial role as a key infrastructure in the architecture of smart cities. The introduction of blockchain into the IoV can improve the performance, scalability, and security of IoV applications. By examining the characteristics of blockchain, which include decentralization, distributed storage, immutability, consensus mechanisms, and smart contracts, researchers have explored its integration with zero-trust architecture to achieve defense-in-depth protection. For example, Garg et al. proposed dynamic micro-segmentation that divides each ECU, OBU, and edge node into isolated security domains, while leveraging blockchain properties to enable fine-grained access control [52].

Recent studies demonstrate advanced applications of this fusion technology. Hussain et al. pioneered blockchain access control schemes using trust–location–risk dimensions, establishing important foundations for zero-trust implementation [97]. Mendiboure et al. developed blockchain-based authentication that maintains data reliability while enabling secure sharing, though the solution requires further latency optimization for real-time vehicle operations [98]. Zhang et al. designed a distributed trust management system that improves malicious vehicle identification, but the consensus algorithm still faces scalability challenges in large-scale networks [99]. Singh et al. subsequently enhanced the framework with three modular components for trust assessment and updates, yet the privacy protection mechanisms need strengthening for sensitive data monitoring scenarios [100].

Emerging research by Su et al. combines federated learning with blockchain to address these gaps, while new challenges are emerging in minimizing verification latency to meet autonomous driving requirements [101]. The field must still resolve critical issues including (1) adaptive access control under dynamic network conditions, (2) balance between decentralization and real-time performance, and (3) lightweight cryptographic primitives for resource-constrained edge nodes.

### 5.3. Digital Twins

The 6G network will bring broader development opportunities for intelligent connected vehicles. With the increase in vehicles and traffic complexity, it has been challenging for traditional traffic management to fulfill the requirements of contemporary traffic. An IoV architecture based on digital twins is an advanced solution that provides more convenient intelligent services by connecting vehicles to the cloud for remote monitoring and control of vehicles. At present, researchers are generally concerned about improving the security of IoV systems. The application of digital twins in IoV security mainly involves the following aspects:

Traffic accident prediction and early warning: Through comprehensive analysis of multi-source data, the digital twin gathers vehicle status and road conditions in the region, senses traffic safety hazards in advance, carries out large-scale multi-dimensional simulations, accurately predicts the accident probability, and sends accurate early-warning information in real time to improve traffic safety.

Virtual driving scenario simulation: Digital twins build multiple virtual simulation environments for autonomous driving algorithm testing and validation. By simulating various traffic scenarios and conditions, they test the reliability and robustness of the automatic driving system and provide development assistance. Through data sensing and mechanism simulation, twin-sensing technology can predict the behavior of intelligent connected vehicles, providing a basis for deeper insight and understanding.

Twin model migration technology: Vehicles are constantly moving, and the service scope and quality of each network edge node are limited. Therefore, for more effectively meeting the requirements of vehicles for low delay and high bandwidth, the digital twin of an intelligent vehicle needs to move with the vehicle, to carry out seamless migration in the network.

In summary, the future trend in IoV security will be the two-way drive of technology and standards; technological innovation will bring improved security protection capabilities. Standardization will ensure coordination and compatibility between different technologies and systems. The challenges lie in maintaining the adaptability of security standards amidst rapidly evolving technology and in effectively integrating various technologies to counter increasingly complex security threats.

### 5.4. Post-Quantum Cryptography for IoV

The rapid advancement of quantum computing capabilities has brought unprecedented security challenges to conventional public-key cryptographic systems, including RSA and ECC algorithms [102]. Within the automotive sector, research teams such as those led by NIST post-quantum cryptography standardization participants have identified the Internet of Vehicles as particularly vulnerable due to its long life-cycle requirements and safety-critical nature. Mao et al. systematically evaluated lattice-based cryptographic constructions, with schemes like CRYSTALS-Dilithium demonstrating superior performance for vehicular security applications [103].

However, research consortiums such as the PQCRYPTO project have identified several unresolved challenges. Primary among these is the tension between security parameters and operational efficiency, while frequent key rotation enhances security, it introduces unacceptable overhead for high-speed vehicle networks. Additional limitations noted by Yang et al. include the need for hardware acceleration to meet real-time requirements in 5G-V2X scenarios, and the lack of standardized implementation guidelines for automotive-grade security modules [104]. The AutoCrypt working group estimates that current lattice-based schemes require approximately 15–20% more computational resources than existing ECDSA implementations when deployed on typical vehicular ECUs.

Ongoing research directions highlighted by the community include the following: hybrid cryptographic approaches combining classical and post-quantum algorithms, optimization of lattice parameter sets for automotive use cases, and development of quantum-resistant key management protocols tailored for distributed vehicular networks. Industry-academic partnerships through organizations like the Connected Vehicle Security Alliance are actively working to address these challenges while maintaining backward compatibility with existing vehicular security infrastructures.

## 6. Conclusions

The rapid advancement of sensing, communication, computing, and intelligent technologies has propelled the IoV into a new era, with ISCCI emerging as a pivotal framework for enhancing collaborative efficiency. However, security challenges across the physical, network, application, and system layers pose significant risks to the IoV’s sustainable development. While this paper has systematically reviewed these threats and proposed countermeasures like physical-layer encryption and blockchain-based authentication, critical gaps remain in real-world validation, quantum-resistant security, cross-layer optimization, and system interoperability. Future research must prioritize these areas by leveraging 6G networks, digital twins, and AI-driven solutions to build resilient security frameworks that can support the full potential of IoV in enabling safe and efficient intelligent transportation systems. Addressing these challenges will be crucial for realizing trustworthy IoV ecosystems in the era of autonomous driving and smart cities.

## Figures and Tables

**Figure 1 sensors-25-05119-f001:**
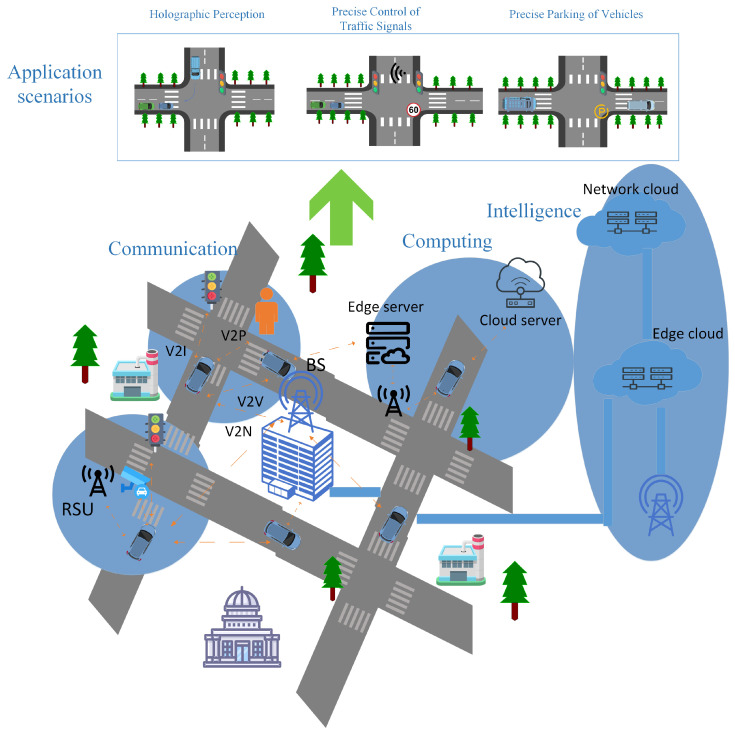
The architecture of the ISCCI.

**Figure 2 sensors-25-05119-f002:**
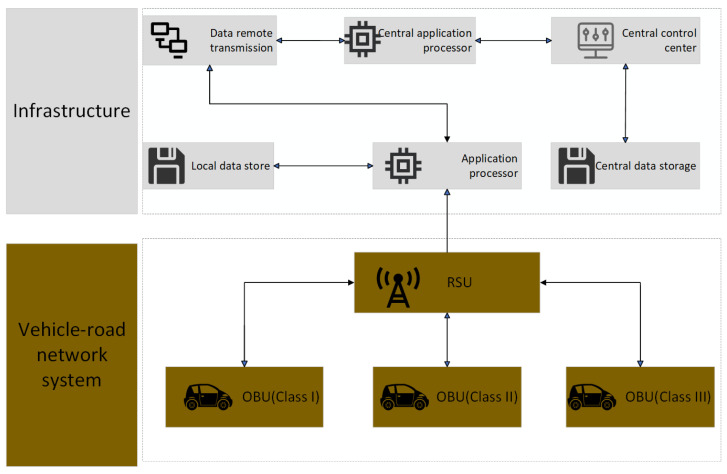
Communication system structure of DSRC vehicle networking scenario.

**Figure 3 sensors-25-05119-f003:**
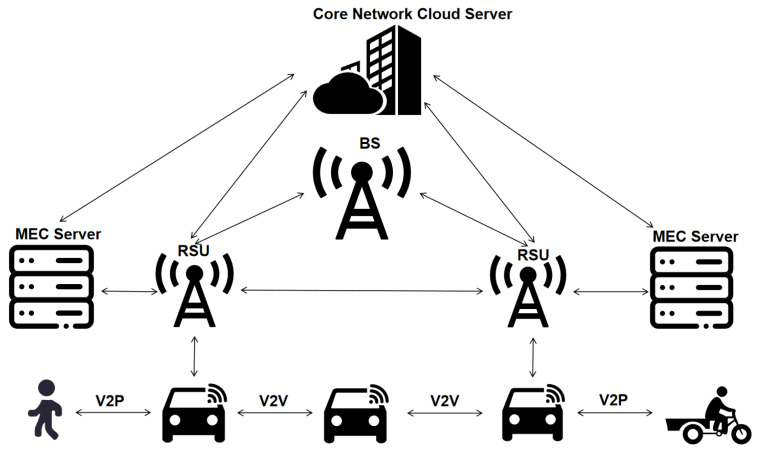
Communication system architecture of C-V2X vehicle networking scenario.

**Figure 4 sensors-25-05119-f004:**
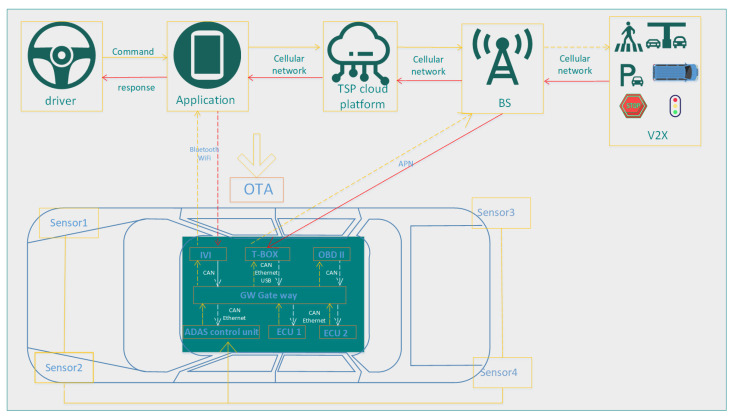
IoV system architecture analysis diagram.

**Figure 5 sensors-25-05119-f005:**
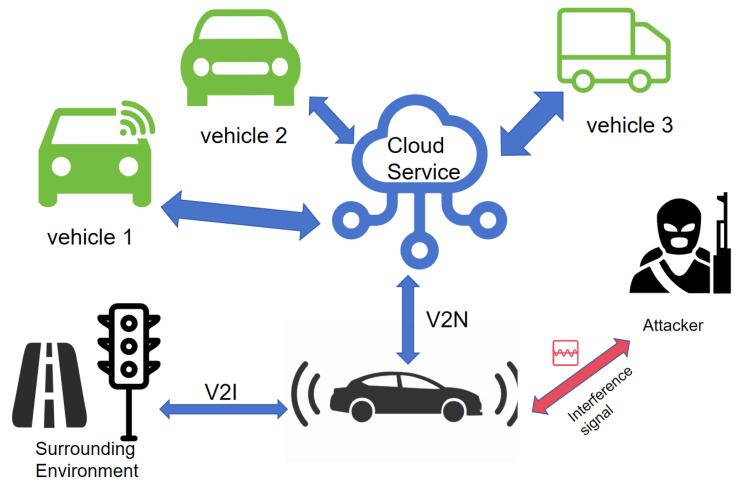
Physical-layer attack diagram.

**Figure 6 sensors-25-05119-f006:**
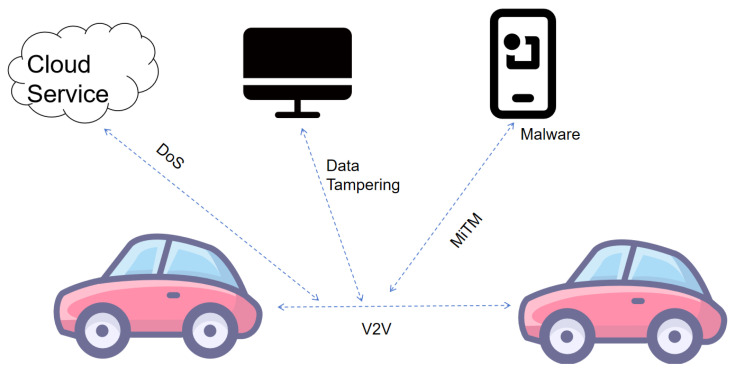
Network-layer attack diagram.

**Figure 7 sensors-25-05119-f007:**
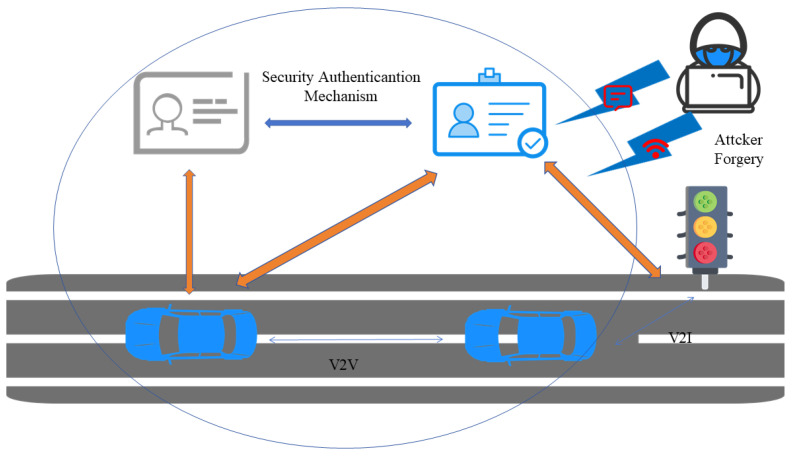
Application-layer attack diagram.

**Figure 8 sensors-25-05119-f008:**
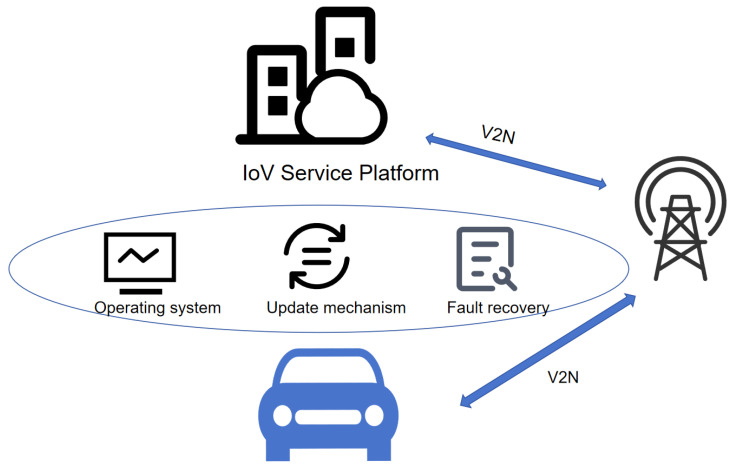
System-layer attack diagram.

**Table 1 sensors-25-05119-t001:** Comparative analysis of ISAC integration methodologies for IoV applications.

Method	Technical Implementation	Operational Scenario	Key Advantages	Major Limitations
OFDM-based ISAC	Adaptive CP-OFDM with radar processing	Urban multi-vehicle networks	94% spectrum utilization 5G NR backward compatibility	High PAPR (12–15 dB) Limited Doppler resolution at >200 km/h
Tunable PAPR ISAC	Adaptive OFDM with peak suppression	High-mobility urban scenarios	PAPR reduced to 6–8 dB Backward-compatible with 5G NR Flexible waveform adaptation	Increased computational complexity Requires real-time optimization
PMCW Radar-Comm	Phase-modulated CW with Gold sequence spreading	Highway platooning	28 dB interference suppression Low probability of intercept/detect	40 MHz instantaneous bandwidth Requires nanosecond-level synchronization
Hybrid Beamforming	64-element mmWave phased array with dual-function RFIC	V2I infrastructure nodes	0.5 m range resolution 0.1° angular accuracy (azimuth/elevation)	15 W typical power consumption 2 ms real-time calibration requirement
Neural-Enhanced ISAC	CNN-LSTM hybrid processing (5.8 TOPS required)	Urban canyon environments	0.3° median AoA/AoD error 5.2 dB SNR improvement over MUSIC	Requires GPU/TPU acceleration 3 ms average inference latency
LEO-Enhanced ISAC	Quantum key distribution (QKD) over Ka-band (26.5–40 GHz)	Cross-regional logistics	99.999% anti-jamming reliability Global coverage capability	12–15 ms inherent satellite link latency High ground station deployment cost

All specifications are experimentally validated per cited references in Section 2.3.

**Table 2 sensors-25-05119-t002:** Performance envelopes of mainstream OBU platforms.

Model	CPU Cores	AI Accelerator	Power (W)	Memory (GB)	Latency (ms)
NVIDIA DRIVE AGX Orin	12× ARM Cortex	2048-core GPU	15–40	32–64	2.1 (YOLOv8)
Qualcomm Snapdragon Ride	8× Kryo	Hexagon DSP	10–30	16–32	3.8 (ResNet50)
Texas Instruments TDA4VM	8× RISC-V	2× C7× DSP	5–20	8–16	5.2 (MobileNetV3)

Measured for 1080 p image processing at 30 FPS. Data sources: NVIDIA (2023), Qualcomm (2022), TI (2022).

**Table 3 sensors-25-05119-t003:** Comprehensive analysis of IoV security challenges and protection strategies.

Security Level	Major Threats	Threat Level	IoV-Specific Impact	Safety Countermeasures	Effectiveness Evaluation	Refs.
Physical Layer	Wireless signal interference Electromagnetic attacks	High Critical	V2X interruption (>200 ms) Disables autonomous driving	Physical-layer encryption Anti-jamming protocols	BER: $10^{-6}$ Detection: 95%	[51,52,53,54,55,56,67,68,69,70,71,72,73,74,75,76,77]
Network Layer	False message injection DoS attacks MiTM attacks S2V link threats	Critical High Medium Critical	Fake traffic signals Brake message disruption Data interception Misguided positioning	Blockchain authentication RL-based detection Secure routing LEO satellite QKD	Auth delay: <50 ms Availability: 85% MITM prev.: 92% Jamming resist.: 99.999%	[57,58,59,60,71,72,73,74,75,76,77]
Application Layer	Authentication flaws Data tampering	Medium High	Unauthorized control Altered diagnostics	Multi-factor auth. Blockchain notarization	Crack diff.: $10^{3}$ times Tamper det.: 100%	[61,62,63,78,79,80,81]
System Layer	OTA attacks Edge compromise	Critical Medium	Mass vehicle hijacking Regional coordination failure	Secure boot chain Zero-trust architecture	Exploit rate: ≤0.1% Lateral mov. stop: 90%	[64,65,66,82,83,84,85,86]

Threat levels: critical (safety critical), high (major functional impact), medium (service quality impact).

## Data Availability

This article has not generated new data, and data sharing is not applicable.

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
