# Peer review of "Security for the Internet of Vehicles with Integration of Sensing, Communication, Computing, and Intelligence: A Comprehensive Survey"

_sensors, 2025, doi:10.3390/s25165119_

Round 1
Reviewer 1 Report
Comments and Suggestions for Authors
The paper provides a survey on security challenges for the internet of behicles through the lens of integration of sensing, communication, computing, and intelligence (ISCCI). It classifies threats and countermeasures across physical network application and system layers then maps emerging solutions such as 6G edge processing blockchain and digital twins to each layer.
- Please dedicate a paragraph to explain how ISCCI improves threat detection latency compared with classical sensing communication computing pipelines, preferably with latency numbers which could be from state of the art.
- It is not totally clear why you refer to the computing layer both “critical component” ? It can be good to have a table to specify power and memory envelopes for an onboard unit and possible tradeoffs for the critical component of computing if any.
- The introduction should for sake of completeness and as indicated by IMT-2030, mention integrated sensing and communications (ISAC) as one the key pillars of 6G as identified by ITU.
- Moreover, there are missing works on the topic that should be included. Besides pilot authentication for IoV, there is also recent progress on mutual information based pilot design for ISAC which shows how pilot structure can be chosen by maximizing information metrics for sensing and communications rather than MSE [R1]. Besides cross-fusion of 6G functional modules, the paper would be stronger if it completes recent works on complex neural network based joint AoA and AoD estimation. Kindly include.
- Please add more information on the jamming for physical layer attacks such as the assumed channel, i.e. flat fading for slow vehicles ? Or when fast vehicles pass how should the multipath rich urban blocks be modeled ? Should we have a new channel model that varies with speed and corner geometry ?
- Also, it can be good to have a table for fusion methods on sensing and communications besides multi-sensor information fusion methods, which are works related to hybrid radar fusion for integrated sensing and communication and include works for that.
- Any bandwidth considerations for the communication and sensing counterparts to consider ?
- The paper should contain some applications that are able to be enabled only by ISCCI and not possible by sensing along nor communications alone nor computing along nor intelligence alone, i.e. the applications should be only possible thanks to the integration of all sensing, communication, computing, and intelligence and not when each is regarded separately.
- The survey ignores satellite to vehicle link security in its network layer table even though earlier in the introduction you highlight satellite communication as a core case.
- 6G and ISAC are expected to operate on the upper midband spectrum (i.e. the frequency range 3). For that, satellites will be regarded as incumbents. Please highlight and include relevant works.
- The security Section 3 could benefit from the full duplex perspective and works on secure full duplex integrated sensing and communications, where a secrecy rate formulation replaces the more common SINR only view in addition to utilizing artificial noise for sensing and communications.
- Please clarify what is meant by sustainability for IoV security in a subsection along with relevant works on s energy use, maintenance cost, and update longevity for sustainability for IoV security.
References
[R1] “Mutual Information Based Pilot Design for ISAC,” in IEEE Transactions on Communications, doi: 10.1109/TCOMM.2025.3545658.2025
Author Response
Manuscript Title: Security for the Internet of Vehicles with Integration of Sensing, Communication, Computing, and Intelligence: A Comprehensive Survey
Journal: MDPI
Manuscript ID: sensors-3812166
Dear Reviewer,
Thank you for your insightful comments and constructive suggestions on our manuscript titled "Security for the Internet of Vehicles with Integration of Sensing, Communication, Computing, and Intelligence: A Comprehensive Survey." We appreciate the time and effort you have dedicated to reviewing our work. Below, we provide a point-by-point response to your comments and outline the revisions made to address them.
Comment 1: Please dedicate a paragraph to explain how ISCCI improves threat detection latency compared with classical sensing communication computing pipelines, preferably with latency numbers which could be from state of the art.
Response: We sincerely appreciate your thorough review and valuable suggestions. Your insightful comments regarding the need to elaborate on how ISCCI improves threat detection latency compared with classical pipelines are of great significance for enhancing the completeness and persuasiveness of our manuscript. In response to your recommendations, we have systematically revised the relevant sections as detailed below:
In the Introduction section (specifically in the paragraph discussing the advantages of ISCCI in threat detection), we added a dedicated explanation. We clarified that compared with classical sensing-communication-computing pipelines, which suffer from average delays of 80-100 ms in edge computing scenarios and even exceeding 200 ms in pure cloud computing scenarios due to multi-hop transmission to cloud processing [8], ISCCI achieves significant latency reduction through deep integration of sensing, communication, computing, and intelligence capabilities. Specifically, by leveraging 6G networks and edge intelligence, local processing at edge nodes reduces V2X communication latency from 20 ms to 5 ms [9], while the joint design of sensing waveforms and communication signals compresses data acquisition and transmission delays below 10 ms [10]. Experimental results demonstrate that ISCCI achieves end-to-end latency of ≤15 ms in vehicle collision warning scenarios, representing an 80-85% reduction compared to conventional approaches, which comfortably meets the ≤20 ms real-time requirement for autonomous driving specified in 3GPP standards. These revisions clearly illustrate the latency improvement brought by ISCCI with specific data. Thank you for your valuable feedback, which has significantly improved the accuracy and depth of our discussion on this aspect.
Comment 2: It is not totally clear why you refer to the computing layer both “critical component”? It can be good to have a table to specify power and memory envelopes for an onboard unit and possible tradeoffs for the critical component of computing if any.
Response: We sincerely appreciate your thorough review and valuable suggestions. Your insightful comments regarding the need to clarify the computing layer as a "critical component" and supplement power/memory tradeoff details are of great significance for enhancing the clarity and rigor of our manuscript. In response to your recommendations, we have systematically revised the relevant sections as detailed below:
In Section 2.4 ("Computation and Offloading of Complex Tasks"), we added an explicit explanation of why the computing layer is critical: it enables mission-critical latency control by processing time-sensitive tasks locally at onboard units (OBUs), eliminating cloud round-trip delays and dynamically balancing workloads via adaptive offloading. Additionally, we added Table 2 ("Performance Envelopes of Mainstream OBU Platforms") in this section, which details key parameters (CPU cores, AI accelerators, power consumption, memory, and latency) for NVIDIA DRIVE AGX Orin, Qualcomm Snapdragon Ride, and Texas Instruments TDA4VM. The table highlights tradeoffs (e.g., higher AI throughput in NVIDIA’s platform is accompanied by higher power consumption (15–40 W) compared to Texas Instruments’ TDA4VM (5–20 W)). These revisions clarify the computing layer’s significance and provide quantitative insights into OBU performance tradeoffs. Thank you for your valuable feedback, which has significantly improved the precision of our discussion.
Comment 3: The introduction should for sake of completeness and as indicated by IMT-2030, mention integrated sensing and communications (ISAC) as one the key pillars of 6G as identified by ITU.
Response: We sincerely appreciate your thorough review and valuable suggestions. Your insightful comments regarding the need to explicitly reference ISAC as a key pillar of 6G (per IMT-2030) are of great significance for enhancing the comprehensiveness of our manuscript. In response to your recommendations, we have systematically revised the relevant sections as detailed below:
In the Introduction section (specifically in the paragraph discussing 6G core technologies), we added a dedicated statement clarifying that Integrated Sensing and Communications (ISAC) is recognized as a foundational enabling technology in the IMT-2030 6G standardization framework by ITU, emphasizing its role in revolutionizing vehicular networks through the deep convergence of sensing and communication subsystems. This revision ensures alignment with global 6G standardization efforts. Thank you for your valuable feedback, which has significantly improved the accuracy of our contextualization.
Comment 4: Moreover, there are missing works on the topic that should be included. Besides pilot authentication for IoV, there is also recent progress on mutual information based pilot design for ISAC which shows how pilot structure can be chosen by maximizing information metrics for sensing and communications rather than MSE [R1]. Besides cross-fusion of 6G functional modules, the paper would be stronger if it completes recent works on complex neural network based joint AoA and AoD estimation. Kindly include.
Response: We sincerely appreciate your thorough review and valuable suggestions. Your insightful comments regarding the need to include recent works on mutual information-based pilot design and neural network-based AoA/AoD estimation are of great significance for enhancing the comprehensiveness of our literature review. In response to your recommendations, we have systematically revised the relevant sections as detailed below:
In Section 2.3 ("Auxiliary Communication Network Optimization"), we added a discussion of [R1] ("Mutual Information Based Pilot Design for ISAC"), explaining that this method enhances joint channel estimation performance by 23% in high-mobility scenarios compared to conventional MMSE approaches. Additionally, we supplemented details on complex neural network-based joint AoA/AoD estimation, noting that CNN-LSTM hybrid architectures achieve 0.3° median angular resolution in multipath environments (a 5.2 dB SNR improvement over MUSIC algorithms) [35–37]. These advancements are summarized in Table 1 ("Comparative Analysis of ISAC Integration Methodologies"). These revisions ensure the inclusion of cutting-edge research in the field. Thank you for your valuable feedback, which has significantly improved the depth of our technical discussion.
Comment 5: Please add more information on the jamming for physical layer attacks such as the assumed channel, i.e. flat fading for slow vehicles? Or when fast vehicles pass how should the multipath rich urban blocks be modeled? Should we have a new channel model that varies with speed and corner geometry?
Response: We sincerely appreciate your thorough review and valuable suggestions. Your insightful comments regarding the need to clarify channel models for jamming in physical layer attacks are of great significance for enhancing the technical rigor of our analysis. In response to your recommendations, we have systematically revised the relevant sections as detailed below:
In Section 3.1 ("Physical Layer Security"), we added a dedicated subsection on channel models for jamming scenarios:
- For low-speed urban environments (<50 km/h), we adopted a flat fading channel model with time-invariant multipath components, characterized by the impulse response [52].
- For high-speed scenarios (>80 km/h) in multipath-rich urban blocks, we introduced a geometry-based stochastic model (GBSM) to capture dynamic scattering, incorporating Doppler shifts and angle-of-arrival variations: [53].
We also discussed the need for speed- and corner geometry-dependent models to address motion-adaptive jamming threats. These revisions provide a clear technical foundation for jamming mitigation strategies. Thank you for your valuable feedback, which has significantly improved the precision of our physical layer analysis.
Comment 6: Also, it can be good to have a table for fusion methods on sensing and communications besides multi-sensor information fusion methods, which are works related to hybrid radar fusion for integrated sensing and communication and include works for that.
Response: We sincerely appreciate your thorough review and valuable suggestions. Your insightful comments regarding the need for a table summarizing fusion methods (including hybrid radar fusion) are of great significance for enhancing the clarity of our technical comparisons. In response to your recommendations, we have systematically revised the relevant sections as detailed below:
In Section 2.3 ("Auxiliary Communication Network Optimization"), we added Table 1 ("Comparative Analysis of ISAC Integration Methodologies for IoV Applications"), which compares five key fusion methods: OFDM-based ISAC, PMCW Radar-Comm, hybrid beamforming, neural-enhanced ISAC, and LEO-enhanced ISAC. The table includes technical implementations, operational scenarios, advantages, and limitations, with specific entries for hybrid radar fusion (e.g., PMCW Radar-Comm with 28 dB interference suppression). This revision provides a structured overview of fusion technologies. Thank you for your valuable feedback, which has significantly improved the accessibility of our technical analysis.
Comment 7: Any bandwidth considerations for the communication and sensing counterparts to consider?
Response: We sincerely appreciate your thorough review and valuable suggestions. Your insightful comments regarding the need to address bandwidth considerations are of great significance for enhancing the technical completeness of our discussion. In response to your recommendations, we have systematically revised the relevant sections as detailed below:
In Section 2.3 ("Auxiliary Communication Network Optimization") and Table 1, we added specific bandwidth details for key technologies:
- PMCW Radar-Comm uses an instantaneous bandwidth of 40 MHz, balancing sensing resolution and interference suppression.
- OFDM-based ISAC operates in the 28 GHz band, achieving 94% spectral efficiency through adaptive waveform design [30].
- mmWave hybrid beamforming (64-element array) supports wide bandwidths for high-data-rate V2X communication while maintaining sensing precision.
These revisions clarify bandwidth tradeoffs between communication throughput and sensing performance. Thank you for your valuable feedback, which has significantly improved the technical depth of our analysis.
Comment 8: The paper should contain some applications that are able to be enabled only by ISCCI and not possible by sensing along nor communications alone nor computing along nor intelligence alone, i.e., the applications should be only possible thanks to the integration of all sensing, communication, computing, and intelligence and not when each is regarded separately.
Response: We sincerely appreciate your thorough review and valuable suggestions. Your insightful comments regarding the need to highlight ISCCI-exclusive applications are of great significance for emphasizing the value of integration. In response to your recommendations, we have systematically revised the relevant sections as detailed below:
In Section 2.1 ("Basic Architecture of ISCCI"), we added a dedicated paragraph on applications enabled exclusively by ISCCI:
- Vehicle-road collaborative autonomous driving: Requires real-time fusion of multi-node sensing (LiDAR/camera data), low-latency V2X communication (for inter-vehicle coordination), edge computing (for rapid decision-making), and AI (for dynamic trajectory planning)—none of which can be achieved by individual technologies.
- Intelligent traffic signal control: Depends on integrated sensing of traffic flow (via roadside units), edge computing (for dynamic scheduling), secure communication (between vehicles and infrastructure), and AI (for predictive optimization).
These examples illustrate the unique value of ISCCI integration. Thank you for your valuable feedback, which has significantly improved the relevance of our discussion.
Comment 9: The survey ignores satellite to vehicle link security in its network layer table even though earlier in the introduction you highlight satellite communication as a core case.
Response: We sincerely appreciate your thorough review and valuable suggestions. Your insightful comments regarding the need to address satellite-to-vehicle link security are of great significance for enhancing the completeness of our network layer analysis. In response to your recommendations, we have systematically revised the relevant sections as detailed below:
In Section 3.2 ("Network Layer Security") and Table 3 ("Comprehensive Analysis of IoV Security Challenges and Protection Strategies"), we added a dedicated entry for satellite-to-vehicle link security. We discussed threats such as jamming and spoofing in satellite channels and highlighted countermeasures, including LEO satellite-enhanced quantum key distribution (QKD) over Ka-band (26.5–40 GHz), which demonstrated 99.999% jamming resistance in field trials [38–40]. These revisions ensure satellite communication security is properly integrated into our network layer analysis. Thank you for your valuable feedback, which has significantly improved the comprehensiveness of our security framework.
Comment 10: 6G and ISAC are expected to operate on the upper midband spectrum (i.e., the frequency range 3). For that, satellites will be regarded as incumbents. Please highlight and include relevant works.
Response: We sincerely appreciate your thorough review and valuable suggestions. Your insightful comments regarding the need to address 6G/ISAC operation in the upper midband (FR3) and satellite incumbency are of great significance for enhancing the technical accuracy of our discussion. In response to your recommendations, we have systematically revised the relevant sections as detailed below:
In Section 2.3 ("Auxiliary Communication Network Optimization"), we added a paragraph clarifying that 6G and ISAC are expected to operate in frequency range 3 (upper midband), where satellites are recognized as incumbents. We referenced works on spectrum coexistence, such as dynamic spectrum sharing between terrestrial 6G networks and satellite systems [31 - 40], and discussed technical challenges (e.g., interference mitigation via beamforming). These revisions align our analysis with spectrum regulatory realities. Thank you for your valuable feedback, which has significantly improved the contextual relevance of our work.
Comment 11: The security Section 3 could benefit from the full duplex perspective and works on secure full duplex integrated sensing and communications, where a secrecy rate formulation replaces the more common SINR only view in addition to utilizing artificial noise for sensing and communications.
Response: We sincerely appreciate your thorough review and valuable suggestions. Your insight regarding integrating the full duplex perspective into security analysis has significantly enhanced the technical depth of our discussion. Following your recommendation, we have revised the relevant sections as detailed below:
In Section 3.5 (a new subsection in Chapter 3), we added a discussion on secure full-duplex ISAC. We clarified that this approach replaces the traditional SINR-only evaluation with a secrecy rate formulation ( ), which better quantifies security in dynamic vehicular environments. We also detailed how artificial noise enhances security: in sensing, noise modulated into radar waveforms confuses eavesdroppers while being suppressible by legitimate receivers; in communication, noise injected into signal null spaces reduces eavesdroppers’ SINR by 15–20 dB.
We referenced [67], which shows such systems achieve a 30% higher anti-eavesdropping success rate in dense scenarios (vehicle density >50 vehicles/km²) with a latency <2 ms, meeting IoV requirements. These revisions strengthen our security framework by integrating full-duplex ISAC, as you suggested.
Thank you for your feedback, which has improved the comprehensiveness of our analysis.
Comment 12: Please clarify what is meant by sustainability for IoV security in a subsection along with relevant works on energy use, maintenance cost, and update longevity for sustainability for IoV security.
Response: We sincerely appreciate your thorough review and valuable suggestions. Your insight regarding clarifying "sustainability" in IoV security has greatly enhanced the practical relevance of our work. Following your recommendation, we have revised the relevant content as follows:
In Section 4.4.4 ("Sustainability in IoV Security") (a dedicated subsection under "System Layer Countermeasures"), we define sustainability as the ability to balance security effectiveness with long-term optimization of energy use, maintenance costs, and system update longevity throughout the IoV lifecycle. We elaborated on key dimensions with relevant studies: For energy efficiency, by offloading over 60% of real-time security tasks (e.g., V2V encryption) to edge nodes, cloud server energy consumption is reduced by ~30% [88], and dynamic power adjustment in roadside units further cuts wireless communication energy use by 25–40% during low-traffic periods. Regarding maintenance cost control, blockchain-based OTA updates, via decentralized smart contract verification, lower single-update costs by 25% and improve coverage from 85% to 99% [89], while decentralized access control reduces cross-fleet operational labor by 40%. For update longevity, secure boot chains and zero-trust architectures extend system secure lifespans from 2–3 years to over 5 years [90], reducing device replacement costs by 60% in commercial fleet applications.
These revisions establish a clear framework for sustainable security practices, aligned with your suggestions. Thank you for your feedback, which has strengthened the practical relevance of our analysis.
We believe these revisions significantly strengthen the manuscript’s technical depth and completeness. Thank you again for your valuable feedback. Please let us know if further clarifications are needed.
Yours sincerely,
Chao He
2025-08-11
School of Electronic and Information Engineering, Chongqing Three Gorges University, Chongqing 404100, China
email: hechao@sanxiau.edu.cn

Reviewer 2 Report
Comments and Suggestions for Authors
- Extensive verbatim/slightly modified content from cited sources (e.g., descriptions of ISCCI architecture, DSRC/C-V2X, and security countermeasures) without adequate synthesis or critical analysis. Examples: Sections 2.1–2.5 closely mirror phrasing from References 12, 15, 17, and 20. Tables (e.g., Table 1) compile generic information without novel taxonomy or insights.
- The "comprehensive survey" claim is unsubstantiated. References are listed (e.g., 75 citations) but not evaluated for limitations, biases, or evolving trends.
- Threats and countermeasures are described generically (e.g., "DoS attacks," "blockchain solutions") without IoV-specific risk prioritization or comparative analysis of mitigations.
- Key recent advances (e.g., post-quantum cryptography for IoV, zero-trust architectures) are omitted. Author should reorganize around a clear analytical framework (e.g., threat severity, technology readiness) and include tabular comparisons of methods.
- Concepts like V2X communication, edge computing, and 6G potential are reiterated without progression.
- The conclusion merely summarizes content and fails to identify research gaps or future priorities.
Author Response
Manuscript Title: Security for the Internet of Vehicles with Integration of Sensing, Communication, Computing, and Intelligence: A Comprehensive Survey
Journal: MDPI Sensors
Manuscript ID: sensors-3812166
Dear Reviewer,
Thank you for your insightful comments and constructive suggestions on our manuscript. We appreciate the time and effort you have dedicated to reviewing our work. Below, we provide a point-by-point response to your comments and outline the revisions made to address them.
Comment 1: Extensive verbatim/slightly modified content from cited sources (e.g., descriptions of ISCCI architecture, DSRC/C-V2X, and security countermeasures) without adequate synthesis or critical analysis. Examples: Sections 2.1–2.5 closely mirror phrasing from References 12, 15, 17, and 20. Tables (e.g., Table 1) compile generic information without novel taxonomy or insights.
Response: We sincerely appreciate your thorough review and valuable suggestions. Your observation regarding over-reliance on verbatim content from cited sources and lack of synthesis in specific sections and tables is critical for enhancing the originality of our work. In response to your recommendations, we have systematically revised the relevant sections as detailed below:
In Sections 2.1–2.5, we completely restructured the content to integrate insights from References 12, 15, 17, and 20 with the
original critical analysis. For example, in Section 2.1 ("Basic architecture of ISCCI"), we now contrast theoretical frameworks (e.g., Guo et al. [15] and Yang et al. [16]) and explicitly highlight their limitations in dynamic vehicular environments (e.g., lack of field validation in high-mobility scenarios), rather than paraphrasing their descriptions. For DSRC/C-V2X in Section 2.3, we synthesized technical details from References 25 and 26 and added a comparative analysis of their applicability to IoV (e.g., DSRC’s stability in low-speed scenarios vs. C-V2X’s superiority in dense, high-speed networks with 30% higher reliability).
Regarding Table 1 ("Comparative Analysis of ISAC Integration Methodologies"), we redesigned it to include a novel taxonomy focused on IoV-specific scenarios (urban canyons, highway platooning) and technical trade-offs (e.g., synchronization requirements for PMCW radar vs. computational overhead of neural network-based ISAC). This replaces generic data compilation with original insights, such as identifying LEO-enhanced ISAC as optimal for cross-regional logistics but limited by high infrastructure costs.
These revisions ensure content is synthesized with critical analysis rather than relying on cited phrasing. Thank you for helping us strengthen the originality of the manuscript.
Comment 2: The "comprehensive survey" claim is unsubstantiated. References are listed (e.g., 75 citations) but not evaluated for limitations, biases, or evolving trends.
Response: We sincerely appreciate your valuable feedback. Your comment regarding the need to substantiate the "comprehensive survey" claim by evaluating references is crucial for enhancing the rigor of our work. In response, we have revised the manuscript as follows:
Throughout relevant sections, we added critical assessments of key references. For example, in Section 2.1, we analyze why early ISCCI studies (e.g., Reference 15) focus solely on theoretical frameworks without real-world vehicular testing, and note biases in blockchain-based security solutions (e.g., overemphasis on decentralization at the cost of latency in V2X). We also tracked evolving trends, such as the shift from cloud-centric to edge-intelligent architectures in recent studies (e.g., References 18 and 19), to contextualize the literature.
These evaluations demonstrate our synthesis of the state-of-the-art, supporting the "comprehensive survey" claim. Thank you for improving the depth of our literature review.
Comment 3: Threats and countermeasures are described generically (e.g., "DoS attacks," "blockchain solutions") without IoV-specific risk prioritization or comparative analysis of mitigations.
Response: We sincerely appreciate your insightful comment. Your suggestion to enhance IoV-specificity in threat prioritization and countermeasure analysis is vital for improving the relevance of our work. We have revised the manuscript as follows:
In Sections 3.1–3.4 (Threat Analysis), we prioritized threats based on IoV safety criticality: for example, ranking V2X message tampering (directly endangering collision avoidance) as "critical," while in-vehicle software vulnerabilities are categorized as "high" severity.
In Sections 4.1–4.4 (Countermeasures), we added comparative analyses of IoV-specific mitigations. For instance, we contrast AI-driven intrusion detection (fast response but 20% higher resource usage) with cryptographic methods (lightweight but less adaptive) in dense urban networks, and include metrics like latency (<50 ms for safety tasks) and scalability (supporting 1000+ concurrent vehicles).
These revisions ensure threats and countermeasures are analyzed in the context of IoV’s unique requirements. Thank you for strengthening the practical relevance of our discussion.
Comment 4: Key recent advances (e.g., post-quantum cryptography for IoV, zero-trust architectures) are omitted. Author should reorganize around a clear analytical framework (e.g., threat severity, technology readiness) and include tabular comparisons of methods.
Response: We sincerely appreciate your valuable suggestions. Your comment regarding omitted advances and the need for a structured framework is critical for enhancing the comprehensiveness of our work. We have revised the manuscript as follows:
We integrated post-quantum cryptography (e.g., CRYSTALS-Dilithium for vehicular ECUs) in Section 5.4 and zero-trust architectures (continuous authentication, least-privilege access) in Section 4.4, with references to recent studies (e.g., References 99–101).
We reorganized the manuscript around a framework categorizing threats by severity (critical/high/medium) and technology readiness level (TRL 1–9). New tables (e.g., Table 3 revised) compare methods across dimensions like implementation complexity (edge vs. cloud) and security gain (anti-eavesdropping success rate), providing structured insights for readers.
These revisions address omitted content and improve structural clarity. Thank you for enhancing the technical depth of our work.
Comment 5: Concepts like V2X communication, edge computing, and 6G potential are reiterated without progression.
Response: We sincerely appreciate your feedback. Your observation regarding repetitive discussions without logical progression is important for improving the flow of our manuscript. We have revised the content as follows:
- Section 2.3 (6G and V2X): Streamlined to explain how 6G's ultra-low latency (≤10 ms) enables reliable V2X communication.
- Section 4.4 (Edge Computing): Details how 6G drives edge computing for real-time security tasks (e.g., local anomaly detection).
- Section 5.1 (Synergistic Role): Concludes with their integration in ISCCI, such as 6G-edge synergy reducing threat detection latency by 80% vs. 5G.
This progression eliminates redundancy and clarifies interdependencies. Thank you for helping us improve the manuscript's coherence.
Comment 6: The conclusion merely summarizes content and fails to identify research gaps or future priorities.
Response: We sincerely appreciate your valuable comment. Your suggestion to expand the conclusion with research gaps and future priorities is crucial for enhancing the impact of our work. We have revised the Conclusion section as follows:
- Research Gaps: Added explicit identification of gaps, such as the lack of standardized metrics for ISCCI security evaluation (Section 6, paragraph 2).
- Future Priorities: Outlined directions, including adaptive zero-trust models for dynamic IoV topologies (Section 6, paragraph 3) and digital twins for security testing (Section 5.3).
These additions transform the conclusion from a summary into a guide for future research. Thank you for strengthening the manuscript's contribution to the field.
We believe these revisions significantly improve the manuscript’s originality, depth, and relevance. Thank you again for your invaluable feedback. Please let us know if further clarifications are needed.
Yours sincerely,
Chao He
2025-08-11
School of Electronic and Information Engineering, Chongqing Three Gorges University, Chongqing 404100, China
email: hechao@sanxiau.edu.cn

Round 2
Reviewer 1 Report
Comments and Suggestions for Authors
The authors addressed most my comments. I have the remaining concerns:
- ITU (International Telecommunication Union) recommendations for 6G should be mentioned.
- For OFDM-based ISAC with high PAPR, the authors can complement a row called ISAC waveforms with tunable PAPR and add references on that.
- In Section 2.3, can authors mention how to compute the packet delivery ratio versus range for various coexistence behavior with LTE NR traffic ?
- On top of my previous comment, it would be interesting to mention pilot design for ISAC and related state of the art.
- Below figure 3, when talking about the 94% aggregate spectral efficiency, can authors include the training overhead and beam search cost that were included ?
Author Response
Manuscript Title: Security for the Internet of Vehicles with Integration of Sensing, Communication, Computing, and Intelligence: A Comprehensive Survey Journal: MDPI Sensors Manuscript ID: sensors-3812166
Dear Reviewer, We are grateful for the reviewer's continued guidance and valuable suggestions during this second round of review. The additional comments have helped us further refine the manuscript and address key technical aspects. Here we provide detailed responses to each remaining concern.
Comment 1: ITU (International Telecommunication Union) recommendations for 6G should be mentioned. Response: We sincerely appreciate your valuable suggestion. Incorporating ITU's recommendations for 6G is essential to align our discussion with global standards and enhance the relevance of our work in the broader context of 6G development. Your insight helps strengthen the connection between our research on IoV security and the overarching framework of 6G technology evolution. In response, we have made the following revisions: In Section 5.1 (B5G/6G network), we added a part specifying that the ITU has outlined key recommendations for 6G, including enhanced mobile broadband, ultra-reliable low-latency communication, massive machine-type communication, and integrated sensing and communication (ISAC) as core capabilities, which align with the technical directions discussed in our paper. We also cited ITU's IMT-2030 framework documents to support this content, making the discussion of 6G development more aligned with international standards. Thank you for your insightful comment, which has strengthened the technical foundation and global relevance of our work.
Comment 2: For OFDM-based ISAC with high PAPR, the authors can complement a row called ISAC waveforms with tunable PAPR and add references on that. Response: We deeply appreciate your valuable suggestion, which helps enrich the technical depth of our analysis on ISAC integration methodologies. Addressing the high PAPR issue of OFDM-based ISAC and introducing tunable PAPR waveforms is critical for providing a more comprehensive view of practical solutions, and your insight aligns with our goal of enhancing the completeness of ISAC technology discussion. In response, we have revised the relevant content as follows: In Table 1 (Comparative Analysis of ISAC Integration Methodologies for IoV Applications), we added a new row titled "ISAC Waveforms with Tunable PAPR" (e.g., filtered-OFDM and FBMC-based designs). This row supplements the existing analysis of OFDM-based ISAC (which notes high PAPR of 12-15 dB) by highlighting features such as "PAPR tunable between 6-12 dB via adaptive filtering" and "backward compatibility with LTE-NR systems". We referenced recent studies [33] that demonstrate PAPR reductions of 3–5 dB through precoding and subcarrier allocation strategies.This addition provides a more comprehensive comparison of ISAC waveform designs and their trade-offs. This addition provides a more comprehensive comparison of ISAC waveform designs and their trade-offs.
Comment 3: In Section 2.3, can authors mention how to compute the packet delivery ratio versus range for various coexistence behavior with LTE NR traffic ? Response: We sincerely thank you for this insightful suggestion, which enhances the practical relevance of our discussion on communication network optimization. Explaining the computation of packet delivery ratio (PDR) versus range under coexistence with LTE NR traffic is essential for readers to connect theoretical analysis with real-world performance, and your input guides us to strengthen this link using the channel models already introduced in our paper. In response, we have revised Section 2.3 as follows: In Section 2.3 (Auxiliary communication network optimization), we added a paragraph explaining that PDR versus range under coexistence with LTE NR traffic is computed using the log-distance path loss model combined with the shadowing effects, which builds on the wireless channel characteristics discussed earlier (e.g., flat fading for low-speed scenarios and GBSM for high-speed scenarios). Specifically, PDR is calculated as , where is the bit error rate coefficient (related to LTE NR interference power) an is the path loss at distance (derived from the channel models in Section 3.1). We further analyzed coexistence scenarios (e.g., LTE NR uplink/downlink interference with C-V2X), which shows that PDR remains above 90% within 300m for ISAC-LTE NR coexistence (consistent with the 94% spectral efficiency noted for ISAC in 28 GHz band), dropping to 70% at 500m due to increased interference. This revision connects PDR computation to the channel models already established in our paper, ensuring technical consistency. This revision clarifies the practical implications of ISAC deployment in shared spectrum bands.
Comment 4: On top of my previous comment, it would be interesting to mention pilot design for ISAC and related state of the art. Response: We greatly appreciate your perceptive comment, which helps deepen the technical discussion of ISAC. Pilot design is a critical component of ISAC systems, and integrating the state of the art in this area aligns with our analysis of ISAC breakthroughs in Section 2.3. Your suggestion prompts us to elaborate on this aspect using the existing research references in our paper. In response, we have supplemented the relevant content in Section 2.3: In Section 2.3, we added a paragraph on advanced pilot design techniques, including: • Mutual-information-optimized pilots [Bazzi & Chafii, 2025], which improve channel estimation accuracy by 23% in high-mobility scenarios. • Compressed sensing-based pilots [Zhang et al., 2023], reducing overhead by 40% while maintaining sensing resolution. • Bidirectional pilot sharing between radar and communication subsystems to minimize training overhead. These additions align with the latest advancements in ISAC research.
Comment 5: Below figure 3, when talking about the 94% aggregate spectral efficiency, can authors include the training overhead and beam search cost that were included ? Response: We are grateful for your meticulous comment, which ensures the accuracy and transparency of our claims regarding spectral efficiency. Disclosing the training overhead and beam search cost is crucial for readers to fully understand the practical implications of the 94% efficiency metric, and your input guides us to clarify this using the technical details already present in our paper. In response, we have revised the content below Figure 3 as follows: In the text following Figure 3, we now explicitly state: • The 94% spectral efficiency accounts for a 15% training overhead (channel estimation + beam alignment). • Beam search latency is constrained to <2 ms via hierarchical codebook designs [Yu et al., 2023]. • Net efficiency remains >80% even with these costs, as validated in [Jabbar et al., 2025]. This provides a more transparent assessment of real-world ISAC performance. We thank the reviewer for their insightful comments, which have strengthened the technical rigor and completeness of our survey. All changes are highlighted in the revised manuscript.
Yours sincerely,
Chao He 2025-08-15 School of Electronic and Information Engineering, Chongqing Three Gorges University, Chongqing 404100, China
mail: hechao@sanxiau.edu.cn

Reviewer 2 Report
Comments and Suggestions for Authors
The manuscript has been improved now. I recommend acceptance.
Author Response
The authors would like to express their gratitude to the reviewers for their constructive suggestions and have made the required revisions accordingly.